# Single domain antibodies against enteric pathogen virulence factors are active as curli fiber fusions on probiotic *E. coli* Nissle 1917

Ilia Gelfat[1,2], Yousuf Aqeel[3], Jacqueline M. Tremblay[4], Justyna J. Jaskiewicz[4], Anishma Shrestha[3], James N. Lee[3], Shenglan Hu[3], Xi Qian[3], Loranne Magoun[3], Abhineet Sheoran[4], Daniela Bedenice[4], Colter Giem[2], Avinash Manjula-Basavanna[2], Amanda R. Pulsifer[3], Hann X. Tu[2], Xiaoli Li[5], Marilyn L. Minus[5], Marcia S. Osburne[3], Saul Tzipori[4], Charles B. Shoemaker[4], John M. Leong[3,6], Neel S. Joshi[2]*

**1** John A. Paulson School of Engineering and Applied Sciences, Harvard University, Allston, Massachusetts, United States of America, **2** Department of Chemistry and Chemical Biology, Northeastern University, Boston, Massachusetts, United States of America, **3** Department of Molecular Biology and Microbiology, Tufts University School of Medicine, Boston, Massachusetts, United States of America, **4** Department of Infectious Disease and Global Health, Cummings School of Veterinary Medicine, Tufts University, North Grafton, Massachusetts, United States of America, **5** Department of Mechanical and Industrial Engineering, Northeastern University, Boston, Massachusetts, United States of America, **6** Stuart B. Levy Center for Integrated Management of Antimicrobial Resistance, Tufts University, Medford, Massachusetts, United States of America

* ne.joshi@northeastern.edu

**Data Availability Statement:** All relevant data are within the manuscript and its Supporting Information files.

## Abstract

Enteric microbial pathogens, including *Escherichia coli*, *Shigella* and *Cryptosporidium* species, take a particularly heavy toll in low-income countries and are highly associated with infant mortality. We describe here a means to display anti-infective agents on the surface of a probiotic bacterium. Because of their stability and versatility, VHHs, the variable domains of camelid heavy-chain-only antibodies, have potential as components of novel agents to treat or prevent enteric infectious disease. We isolated and characterized VHHs targeting several enteropathogenic *E. coli* (EPEC) virulence factors: flagellin (Fla), which is required for bacterial motility and promotes colonization; both intimin and the translocated intimin receptor (Tir), which together play key roles in attachment to enterocytes; and *E. coli* secreted protein A (EspA), an essential component of the type III secretion system (T3SS) that is required for virulence. Several VHHs that recognize Fla, intimin, or Tir blocked function *in vitro*. The probiotic strain *E. coli* Nissle 1917 (EcN) produces on the bacterial surface curli fibers, which are the major proteinaceous component of *E. coli* biofilms. A subset of Fla-, intimin-, or Tir-binding VHHs, as well as VHHs that recognize either a T3SS of another important bacterial pathogen (*Shigella flexneri*), a soluble bacterial toxin (Shiga toxin or *Clostridioides difficile* toxin TcdA), or a major surface antigen of an important eukaryotic pathogen (*Cryptosporidium parvum*) were fused to CsgA, the major curli fiber subunit. Scanning electron micrographs indicated CsgA-VHH fusions were assembled into curli fibers on the EcN surface, and Congo Red binding indicated that these recombinant curli fibers were produced at high levels. Ectopic production of these VHHs conferred on EcN the cognate binding activity and, in the case of anti-Shiga toxin, was neutralizing. Taken together, these

**Funding:** This work was supported by the National Institutes of Health (1R01DK110770-01, R01DK113599-01) and the Bill and Melinda Gates Foundation (OPP1172434) (CBS/JML). SH received support from the Program of Study Abroad for Innovative Talents of Science and Technology by Guangdong Academy of Agricultural Science. The funders had no role in study design, data collection and analysis, decision to publish, or preparation of the manuscript.

**Competing interests:** The authors have declared that no competing interests exist.

results demonstrate the potential of the curli-based pathogen sequestration strategy described herein and contribute to the development of novel VHH-based gut therapeutics.

## Author summary

Enteric pathogens are the causative agents of diarrheal disease–a leading cause of infant morbidity and mortality worldwide. While treatment and prevention options such as drugs or vaccines exist for some pathogens, their efficacy and availability are often limited. New therapeutic strategies are therefore needed, especially inexpensive agents in low-income countries where enteric disease burdens are highest. One promising avenue for novel treatments uses VHHs–highly stable, well-expressed, antibody domains derived from camelid species such as llamas and alpacas. The small size, high stability and simple structure of these antibody fragments enables their streamlined production by bacteria such as *E. coli*, potentially reducing cost and improving scalability. In this work, we describe the development of VHHs targeting multiple virulence factor proteins of pathogenic *E. coli* and other leading causes of diarrheal disease. These VHHs provide new tools for the research community and may serve as promising components of agents that prevent or treat pathogen infections. Towards that goal, we engineered a novel system in which the probiotic bacterial strain *E. coli* Nissle 1917 (EcN) is used to express and display VHHs at high density on its surface. By demonstrating the ability of these engineered EcN to bind to pathogens, we provide a first step toward using such probiotics as a cheap, simple, and effective treatment for enteric pathogen infections.

## Introduction

Enteric pathogens, which include viruses, bacteria, and eukaryotic microbes, are a major cause of global morbidity and mortality. These pathogens take a particularly heavy toll in low-income countries where diarrheal disease remains a major cause of infant mortality [1,2]. Traditional interventions such as antibiotics and vaccines suffer from limited efficacy, distribution and implementation challenges, and the rise of antimicrobial resistance [3]. Virulence factors have been identified for many important enteric microbes, but conventional measures to prevent or treat diarrheal disease based on these factors have proved difficult to develop. Therefore, new therapeutic strategies are needed.

One of the leading causes of infant diarrheal disease and associated mortality in low- and middle-income countries is enteropathogenic *Escherichia coli* (EPEC) [1,4–6]. Colonization by EPEC is facilitated by flagella- (Fla-) driven motility that promotes penetration of the mucus layer and association with the intestinal epithelium, where bacteria induce the formation of 'attaching and effacing' (AE) lesions [7,8]. These lesions, which enable epithelial colonization, are characterized by the effacement of microvilli and the induction of filamentous actin 'pedestals' beneath bacteria closely associated with intestinal epithelial cells [9,10]. To generate AE lesions, EPEC utilizes a type III secretion system (T3SS) to translocate the bacterial effector Tir (translocated intimin receptor) into host cells, where it localizes to the plasma membrane and binds to the EPEC surface adhesin intimin. Intimin-mediated clustering of Tir triggers the assembly of filamentous actin beneath bound bacteria. The related pathogen, Shiga toxin-producing enterohemorrhagic *E. coli* (EHEC), a food-borne pathogen which causes systemic illness in high-income regions such as the U.S. and Europe, generates AE lesions by a similar

mechanism [9,11–13], as do some veterinary pathogens such as rabbit enteropathogenic *E. coli* (REPEC) and the mouse pathogen *Citrobacter rodentium* [14–16].

The direct administration of antibodies or antibody fragments has been proposed as a potential treatment for enteric diseases of diverse etiology [17–23]. VHHs, the variable domain of camelid heavy-chain-only antibodies (also known as 'nanobodies'), appear particularly well suited for this application [18,24–28]. Unlike conventional antibodies, VHH antibodies can be efficiently and functionally expressed in *E. coli* thanks to their small size and single-domain structure. Furthermore, VHHs are effectively expressed as fusion proteins with other VHHs, thus potentially enhancing avidity, increasing specificity, and enabling binding to multiple targets [29]. Fusion with other functional domains also adds further versatility to their use as therapeutic agents. Together, these properties confer the potential to reduce production costs, improve scalability, and enable novel therapeutic applications.

Although VHHs have opened many novel therapeutic avenues, several challenges remain for their implementation as intestinal therapeutics. Despite the inherent stability and robustness of many VHHs [30], the harsh chemical and enzymatic conditions and continuous flow found in the GI tract will likely promote the degradation and clearance of VHHs before they reach their target. The delivery of sufficient, stable, and functional VHHs to the gut environment therefore constitutes a substantial hurdle. Additionally, producing, purifying, and formulating large amounts of VHHs is likely to be resource- and labor-intensive, effectively limiting the practicality of such approaches, a particularly relevant limitation for implementation in low-income nation where the enteric disease burden is highest.

Engineered living therapeutics are an alternative strategy for localized production and delivery of molecules to the gut. By genetically modifying a suitable nonpathogenic bacterial strain, heterologous proteins of interest can be produced *in situ*, circumventing the challenges associated with traditional drug delivery strategies [31–33]. The ability to utilize bacteria as a therapeutic agent, bypassing the need for protein purification, can potentially render engineered living therapeutics inexpensive and scalable. *E. coli* Nissle 1917 (EcN) has emerged in recent years as a leading candidate for such approaches [34,35]. EcN has an excellent track record of safety through decades of use as a probiotic, and has also been shown to reduce the severity of ulcerative colitis symptoms [36], as well as interfere with the pathogenicity of several enteric pathogens [37], in part due to its ability to colonize the human gastrointestinal tract [38,39]. Transient colonization of humans has also been shown using engineered EcN [40]. Notably, EcN and other laboratory *E. coli* strains can produce VHHs, as demonstrated by numerous studies [41,42].

Curli fibers are the main proteinaceous components of *E. coli* biofilms. In previous work, we used engineered EcN to display modified curli fibers *in vivo* [43]. By fusing heterologous protein domains to CsgA–the major curli subunit–we were able to construct a cell-anchored mesh of robust protein fibers endowed with novel functionalities, ranging from the display of anti-inflammatory peptides [43] to the nucleation of gold nanoparticles [44]. By fusing pathogen surface-binding VHHs to CsgA, we sought to adapt this strategy to enable EcN to bind enteric pathogens *in situ*, thereby interfering with their pathogen-host interactions and possibly resulting in pathogen elimination. We call this approach "curli-based pathogen sequestration", drawing an analogy to the polymer sequestrants used to remove excess ions from the gut in chronic kidney disease and a handful of other disorders [45,46].

Here, we describe the generation and characterization of novel VHHs targeting the Fla, Tir, intimin and EspA antigens of several EPEC, REPEC, EHEC, and *Citrobacter* strains. We then fused a subset of these VHHs, along with several previously described VHHs that bind virulence factors from other enteric pathogens, to CsgA. By expressing these modified curli fibers in EcN and testing their function, we demonstrated the efficacy of the curli-based

sequestration approach *in vitro* against several pathogenic *E. coli* virulence factors. Finally, we showed that EcN producing CsgA-VHH fusions are capable of recognizing surface antigens on two other major enteric pathogens, *Shigella flexneri* and the eukaryotic pathogen *Cryptosporidium parvum*.

## Results

### Generation and initial characterization of VHHs that recognize Fla, Tir, intimin, or EspA

With the goal of obtaining VHHs that bind to selected virulence factors of AE members of the pathogenic *E. coli* family (Fig 1), we immunized alpacas with these virulence factor antigens, either purified from selected enteric pathogens or prepared as recombinant proteins. For anti-flagella VHHs, we immunized with both purified REPEC or EPEC flagella or purified recombinant FliC proteins from multiple pathogenic *E. coli* species. Recombinant REPEC EspA, EHEC and *C. rodentium* intimins, and EHEC Tir protein were also employed as additional virulence factor immunogens. Unlike flagella, the EspA, Tir, and portions of the intimin proteins are relatively well conserved [47], increasing the likelihood of identifying VHHs that recognize diverse AE pathogens. Following immunization, phage-displayed VHHs prepared from the alpaca B cells were panned and then screened for binding to the immunizing antigens and to orthologous antigens from related AE pathogens as described in Materials and Methods. VHH DNA sequences were then determined, and one or two representative VHHs from each different VHH family (i.e., a family constitutes those VHHs apparently derived from a common B cell progenitor) were selected for soluble protein expression. The selected VHHs and their binding properties are summarized in Table 1. Enzyme-linked immunosorbent assays (ELISAs) were used to estimate the apparent affinity (EC$_{50}$ value) of the VHHs for their original antigen, as well as their apparent affinity for homologous targets from other AE pathogens. VHHs varied widely with regard to their binding capacities and cross-specificities (Table 1

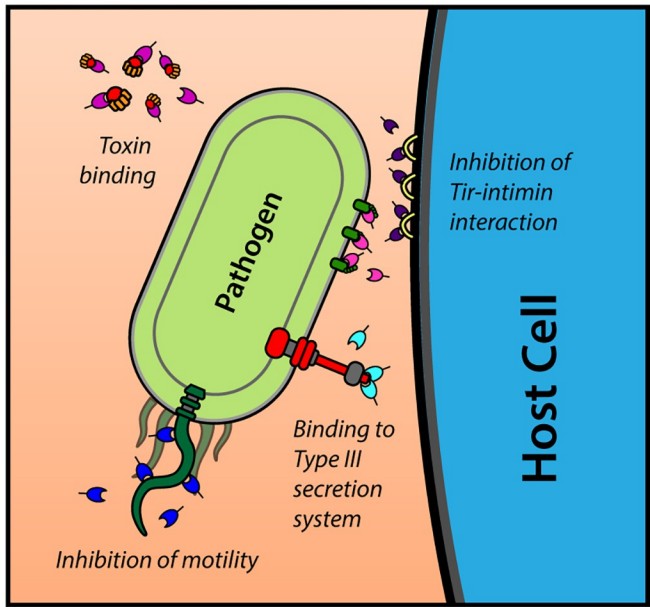

**Pathogenic Virulence Factors**                    **Inhibition by Soluble VHHs**

**Fig 1. Schematic overview of bacterial virulence factors used as VHH targets in this study.**

and S1 Fig). Note that some of the selected VHHs were assigned a simplified name related to their target antigen, e.g., "αInt-12" (i.e., "anti-intimin 12").

Despite our success in identifying several VHHs that recognized purified intimin proteins from various pathogenic *E. coli*, each was highly specific to intimin from only one species, and, importantly, none recognized *E. coli* strain MC1061 expressing EPEC or EHEC intimin on the bacterial surface (e.g., αInt-13, Table 1). Since VHHs are particularly dependent on conformational epitopes [48], we hypothesized that, when coated onto plastic wells, recombinant intimins may not always mimic the conformation of intimin as displayed by *E. coli* [49,50], in spite of demonstrated function in other assays [51]. To test this idea, we used *E. coli* K12 strain MC1061 expressing EHEC or EPEC intimin as an antigen to select and identify phage-displayed VHHs that recognize bacterial surface intimins. Using this process, we discovered VHHs αInt-12, -14 and -17, which recognized both EPEC and EHEC intimin expressed on the surface of strain MC1061, but did not recognize non-intimin producing MC1061 or recombinant intimin (Table 1 and S1 Fig). These data highlight the importance of using conformationally native antigens in preference to recombinant proteins for identifying VHHs of interest.

## Anti-Fla VHHs inhibit REPEC motility

Flagellin (FliC) is the major protein of flagella, which are required for motility and colonization of some pathogenic *E. coli* [52]. We found that REPEC bacteria were highly motile when applied onto low percentage agar plates (Fig 2A). We therefore utilized this strain to test whether various anti-Fla VHHs could impair motility. REPEC was grown to mid-log phase and incubated with either PBS or different concentrations of VHHs (as described in Materials and Methods). After depositing bacteria onto the center of an agar plate, plates were incubated at room temperature for 24h and REPEC motility was then assessed by measuring the colony diameter. Fig 2A shows that αFla-1 VHH did not inhibit motility, whereas motility was inhibited by αFla-6. Two other Fla VHHs, αFla-3 and 4 also showed marked inhibition of REPEC motility at 6.1 and 2.4 μM concentrations (Fig 2B), with an $EC_{50}$ as low as 1.0 μM (Table 1), whereas αFla-5 showed no motility inhibition, even at 6.1 μM (Fig 2B).

## Specific anti-Tir VHHs block the interaction between intimin and Tir

The interaction between intimin and Tir is required for bacterial infection of the host. To determine if anti-Tir VHHs can block this interaction, ELISA-based assays were performed. Plates were coated with GST-tagged EPEC Tir, then incubated with VHHs or (as a negative control) 0.1% BSA. Wells were incubated with either GST-tagged EPEC intimin, or with GST alone (as a negative control), and bound GST or GST-intimin was detected with anti-GST antibody. As shown in Fig 3A, 0.1% BSA showed no ability to block the intimin-Tir interaction, whereas six out of fourteen anti-Tir VHHs showed significant neutralization activity, with αTir-2, αTir-8 and αTir-14 displaying the most potent activity. Thus, despite high affinity for soluble Tir, many anti-Tir VHHs were ineffective in blocking the interaction between purified Tir and intimin *in vitro*.

## EPEC pedestal formation is inhibited by anti-Tir and anti-intimin VHHs

During EPEC infection, bacterial Tir protein on host cells binds to intimin on the bacterial surface. This clustering promotes F-actin assembly beneath bacteria bound at the host plasma membrane and results in the formation of actin pedestals that facilitate pathogen colonization [9]. As αTir-2 was shown to block the intimin-Tir interaction in an ELISA-based assay, we tested whether it was also able to inhibit EPEC pedestal formation. HeLa cells were incubated with EPEC and 100 nM αTir-2 VHH for 3 h at 37˚C, then stained with DAPI (to stain cell

**Table 1. Selected VHHs.**

| (a) Anti-Fla VHHs | | | | | | | | |
|---|---|---|---|---|---|---|---|---|
| VHH name | Vector name | Simplified name | Immunogen | Panned on | EC$_{50}$ [a] | | | REPEC motility inhibition |
| | | | | | REPEC flagella | EPEC flagella | EPEC FliC | |
| JUV-B11 | JVE-2 | αFla-1 | REPEC flagella; EPEC rFliC | REPEC flagella | Trace [b] | NB | Trace | - |
| JUV-C4 | JVE-4 | αFla-2 | | | 0.5 | 1 | 0.3 | ND |
| JUV-E8 | JVE-5 | αFla-3 | | | 0.5 | Trace | NB | + |
| JUV-G8 | JVE-7 | αFla-4 | | | 5 | NB | 10 | + |
| JUV-H1 | JVE-10 | αFla-5 | | | NB | NB | NB | - |
| JUV-H5 | JVE-11 | αFla-6 | | | 5 | NB | 25 | + |
| JWU-F3 | JXA-1 | | | MC1061/EPEC intimin | 10 | 0.2 | ND | ND |
| JWU-H4 | JXA-5 | | | | 3 | 10 | ND | ND |
| JXE-B1 | JXK-1 | | EPEC flagella | EPEC flagella | NB | 0.1 | ND | ND |

| (b) Anti-EspA VHHs | | | | | | | |
|---|---|---|---|---|---|---|---|
| VHH name | Vector name | Simplified name | Immunogen | Panned on | EC$_{50}$ [a] | | Pedestal blocking activity |
| | | | | | REPEC EspA | *C. rodentium* EspA | |
| JXF-D7 | JXM-6 | αEspA-1 | REPEC EspA | REPEC EspA | 0.7 | 0.7 | - |
| JXF-D8 | JXM-8 | αEspA-2 | | | 0.7 | 0.7 | - |
| JXF-H9 | JXM-12 | αEspA-3 | | | 0.7 | 0.7 | - |
| JXF-C4 | JXM-15 | αEspA-4 | | | Trace | 0.7 | - |
| JYB-B1 | JYE-1 | αEspA-5 | | αEspA-1-captured REPEC EspA [c] | 0.3 | 0.3 | ND |
| JYB-B8 | JYE-2 | αEspA-6 | | | 0.7 | 0.7 | ND |
| JYB-D1 | JYE-3 | αEspA-7 | | | 3 | 3 | ND |
| JYB-H4 | JYE-4 | αEspA-8 | | | 0.5 | 0.5 | ND |
| JYB-H6 | JYE-5 | αEspA-9 | | | 0.3 | 0.3 | ND |

| (c) Anti-Tir VHHs | | | | | | | | |
|---|---|---|---|---|---|---|---|---|
| VHH name | Vector name | Simplified name | Immunogen | Panned on | EC$_{50}$ [a] | | | Tir-intimin blocking activity [d] |
| | | | | | EHEC Tir | EPEC Tir | REPEC Tir | |
| JVB-C6 | JVG-1 | αTir-1 | EHEC Tir | REPEC Tir | trace | 50 | trace | - |
| JVB-G4 | JVG-2 | αTir-2 | | | 0.1 | 0.1 | 0.1 | ++ [e] |
| JVB-G8 | JVG-3 | αTir-3 | | | 0.2 | 0.2 | 0.2 | - |
| JVC-C6 | JVI-1 | αTir-4 | | EHEC Tir | 0.1 | 0.2 | 0.1 | + |
| JVC-D10 | JVI-2 | αTir-5 | | | 0.1 | 0.2 | 0.2 | - |
| JVC-E5 | JVI-3 | αTir-6 | | | 0.1 | 0.2 | 0.2 | + |
| JVA-A1 | JVF-1 | αTir-7 | | | 0.7 | 10 | 3 | - |
| JVA-C8 | JVF-2 | αTir-8 | | | 0.1 | 0.2 | 0.2 | +++ |
| JVA-C9 | JVF-3 | αTir-9 | | | 0.1 | 5 | 0.2 | - |
| JVA-D4 | JVF-4 | αTir-10 | | | 0.5 | 0.5 | 0.5 | - |
| JVA-F6 | JVF-7 | αTir-11 | | | 0.1 | 25 | 0.2 | - |
| JVA-D11 | JVF-8 | αTir-12 | | | 0.2 | 0.2 | 0.2 | + |
| JVA-E10 | JVF-12 | αTir-13 | | | 0.5 | 0.2 | 0.1 | - |
| JVA-G1 | JVF-14 | αTir-14 | | | 0.1 | 0.2 | 0.2 | +++ |

| (d) Anti-intimin VHHs | | | | | | | | |
|---|---|---|---|---|---|---|---|---|
| VHH name | Vector name | Simplified name | Immunogen | Panned on | EC$_{50}$ [a] | | | Pedestal blocking activity [f] |
| | | | | | EHEC intimin | EPEC intimin | MC1061 /EPEC intimin | |
| JWS-H4 | JWZ-5 | αInt-12 | EHEC intimin | *E. coli* 1061/pInt (EHEC) | Trace | Trace | 10 | - |
| JWT-C1 | JWZ-7 | αInt-13 | | | 0.5 | Trace | NB | + |
| JWU-D8 | JWZ-9 | αInt-14 | | *E. coli* 1061/pInt (EPEC) | Trace | Trace | 0.5 | + |
| JWU-G8 | JWZ-15 | αInt-17 | | | NB | NB | 0.5 | + |

(*Continued*)

**Table 1.** (Continued)

| JXN-E2[g] | JXS-2 | | *C. rodentium* intimin | DH5α/pInt (*C. rodentium*) | NB | NB | NB | ND |
|---|---|---|---|---|---|---|---|---|

[a] EC$_{50}$ estimates based on dilution ELISAs such as shown in S1 Fig

[b] Trace–EC$_{50}$ >125 nM, i.e., poor but detectable

[c] Panning employed JXF-D7-captured REPEC EspA target

[d] Tir-intimin binding inhibition from Fig 3: + p<0.01; ++ p<0.001; +++ p<0.0001

[e] αTir-2 also displayed pedestal blocking activity, Fig 3

[f] Pedestal blocking activity from Fig 3

[g] JXN-E2 binds to *C. rodentium* intimin with EC$_{50}$ ~0.5 nM

NB–no binding; ND–not done

nuclei) and phalloidin (to stain F-actin). Coded images of each sample were scored blindly on a scale of 1 to 5 (see Materials and Methods), where a score of 5 indicates many pedestals. As expected, EPEC infection in the absence of any VHH resulted in robust pedestal formation (Fig 3B), as did the addition of a negative control VHH (αFla-2) (Fig 3C). These two controls each yielded a pedestal score of 5. However, 20 nM αTir-2 partially inhibited pedestal formation (Fig 3D), and 100 nM αTir-2 completely inhibited EPEC pedestal formation, with no observable pedestals, (Fig 3E). Thus, αTir-2 was shown to inhibit *in vitro* both the interaction between recombinant Tir and intimin, and EPEC pedestal formation, which is triggered by the interaction of these two virulence factors.

Anti-intimin VHHs were also tested for the ability to inhibit EPEC pedestal formation. While αInt-12 showed no inhibition of pedestal formation (Fig 3F and 3G), αInt-13 and -14 both displayed inhibitory activity (Fig 3H and 3I).

Finally, we tested the ability of αEspA-1, 2, 3, and 4 to block pedestal formation (αEspA-1, 2 and 3 are closely related). Despite their high-affinity binding, none of these anti-EspA VHHs were able to inhibit pedestal formation (S2 Fig). However, a panel of anti-EspA VHHs recognizing a different, non-competing epitope were later isolated by panning the library on αEspA-1 VHH-captured EspA (Tables 1 and S2, VHHs of the JYB series). This new panel will be tested for neutralization of pedestal formation in the future.

### Design and expression of curli-VHH fusions in engineered EcN

Next, we chose 12 anti-enteric pathogen VHHs with which to construct CsgA fusions and test the incorporation of various CsgA-VHH on protein fibers produced by EcN while preserving VHH function. This was motivated in part by our previous work, which utilized these robust extracellular fibers to display therapeutic domains *in vivo* [43]. We aimed to harness curli-based display in a similar fashion, to create a multivalent pathogen-binding matrix, which can be delivered and produced within the gut environment by a self-renewing population of engineered microbes.

The VHHs we selected included several anti-virulence VHHs described above as well as VHHs neutralizing enteric toxins TcdA or Stx2, and VHHs binding *Shigella* or *Cryptosporidium* surface antigens (Table 2). The CsgA and VHH coding sequences were fused in-frame and inserted into expression plasmid pL6FO, in which a synthetic version of the full curli operon (*csgBACEFG*) was placed under control of an inducible promoter (Fig 4A). The two protein domains were connected by a 14 amino acid glycine-serine linker, and a 6xHis-tag was appended to the C-terminus of the VHH domain to facilitate detection. To eliminate potential confounding effects of native CsgA, the plasmids were introduced into EcN strain PBP8, in

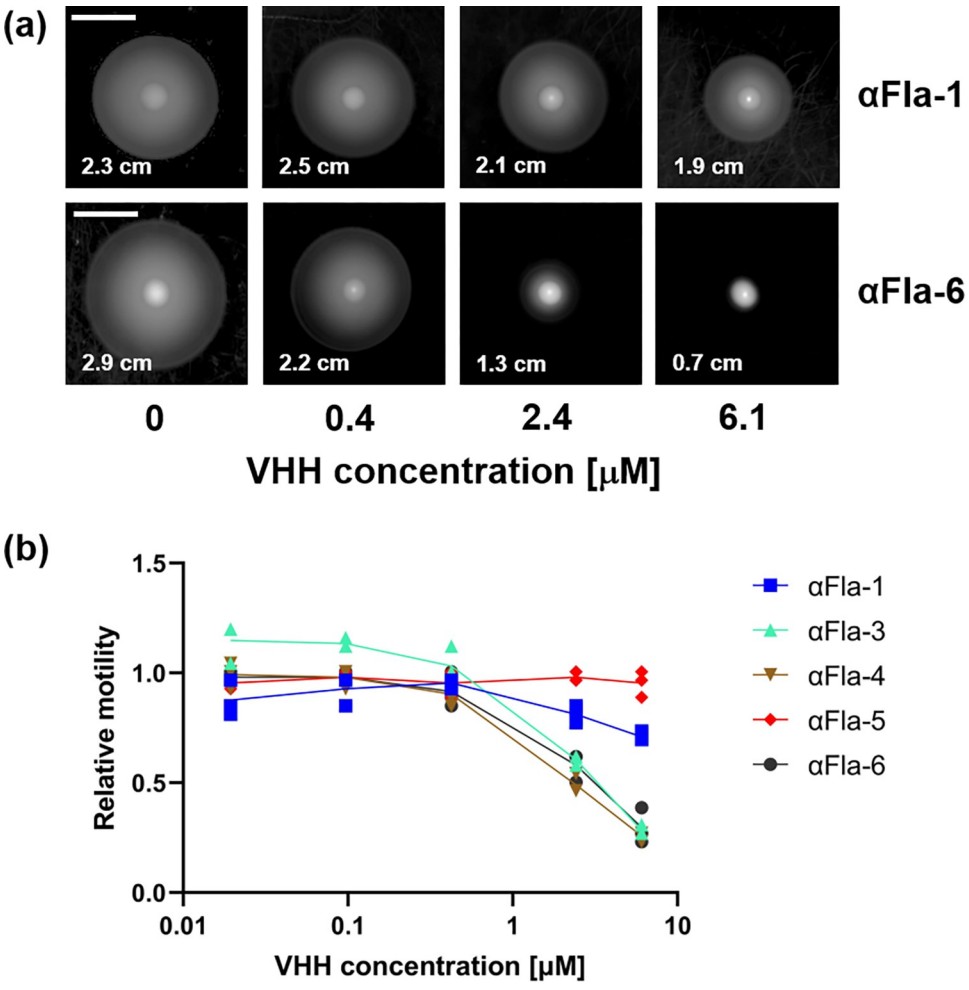

**Fig 2. Anti-Fla VHHs can inhibit REPEC motility.** (a) Representative images of REPEC growth on motility plates after incubation with varying VHH concentrations with spread diameter indicated (scale bar = 1 cm). (b) Relative motility as a function of αFla VHH concentration. Spread diameters were normalized to a "no VHH" control.

which the native curli operon was deleted from the genome [43,53]. We provide an idealized schematic of curli fiber assembly to help explain the incorporation of CsgA-VHH fusions into fibers (Fig 4B).

Following transformation and induction, curli expression and fiber formation were assessed by a Congo Red binding assay, which detects curli on the surface of bacteria [44,54]. Although the degree of Congo Red binding varied slightly between different VHH fusions, overall signal remained high compared to the negative control condition, i.e., plasmid-free strain PBP8 containing neither the wild-type *csgA* gene nor any curli fusion (Fig 4D). This result therefore indicates the formation of cell-anchored protein fibers on the CsgA-VHH-expressing strains. We also used a previously reported curli-specific assay that relies on a shift in Congo Red fluorescence upon binding to curli fibers that can be measured in growing bacterial cultures to compare assembly kinetics between two of the CsgA-VHH variants and PBP8 secreting unmodified CsgA [55]. These experiments demonstrated that CsgA-αFla3 and CsgA-αgp900-2 can form curli fibers at rates comparable to unmodified CsgA when produced by PBP8 (S7 Fig). To further characterize the fusion proteins, field emission scanning electron microscopy images of CsgA-VHH fibers were captured. All CsgA-VHH constructs tested

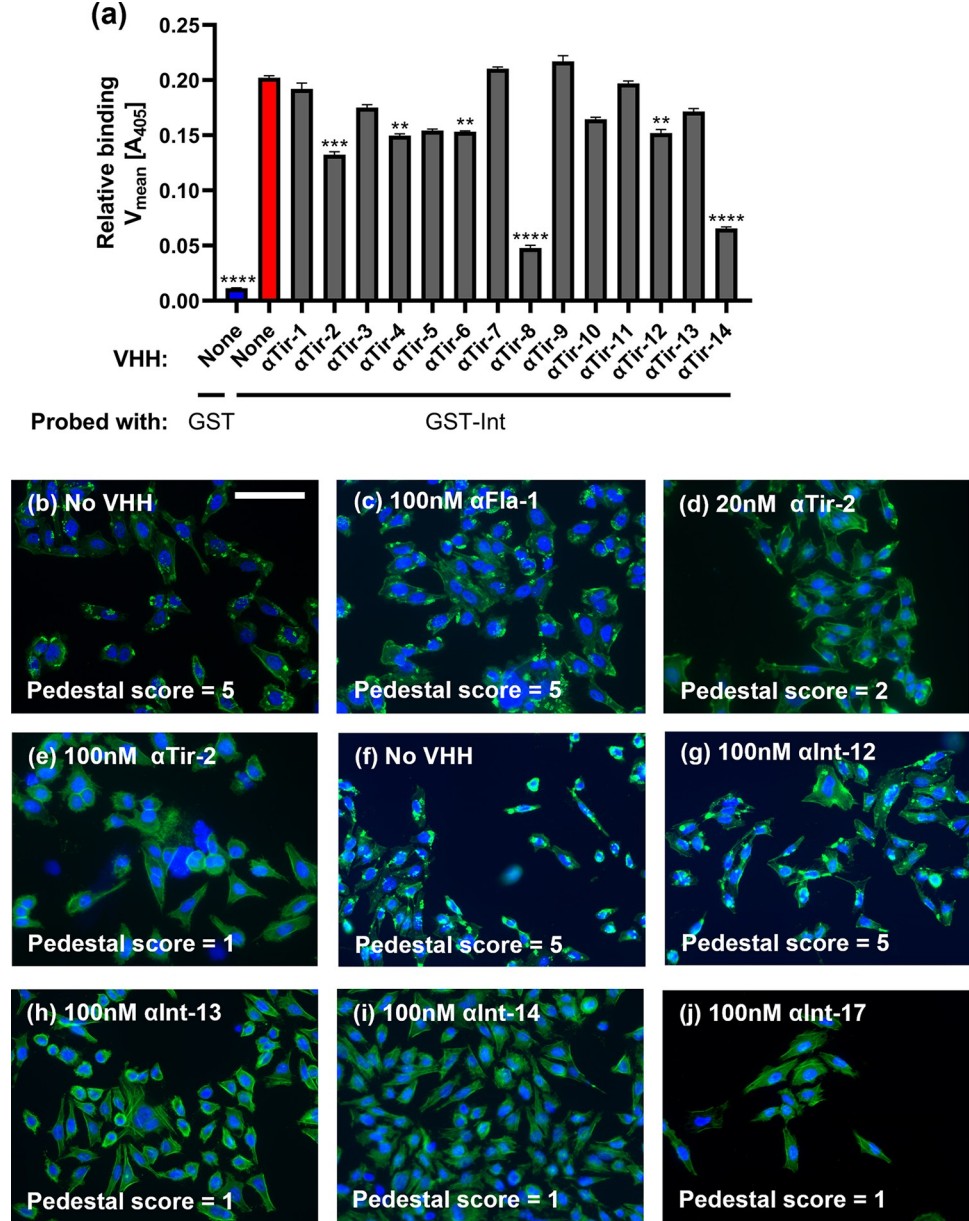

**Fig 3. Anti-Tir and anti-intimin VHHs interfere with Tir-intimin binding and pedestal formation.** (a) Several anti-Tir VHHs blocked Tir-intimin binding. Data presented as mean ± SEM. P-values calculated by one-way ANOVA. ** P < 0.01; *** P < 0.001, **** P < 0.0001. (b-j) Anti-Tir and anti-intimin VHHs can inhibit EPEC-mediated pedestal formation (scale bar = 100 μm). HeLa cells were exposed to EPEC incubated with VHH, fixed, and stained with DAPI (blue) and Alexa Fluor-488 Phalloidin (green). Blinded pedestal scores were assigned by a separate researcher.

resulted in the formation of curli material with morphology similar to the wild-type curli fibers, as produced and assembled by PBP8 based on previously published results (Figs 4C and S3) [43]. Wide angle X-ray scattering (WAXS) of dried films composed of two of the CsgA-VHH fiber networks (CsgA-αFla3 and CsgA-αgp900-2) showed a characteristic d-spacing peak at 0.44–0.45 nm, identical to films composed of CsgA-6xHis (S8 Fig). This result is also in agreement with previous reports in the literature for different CsgA fusion proteins

**Table 2.  CsgA-VHH constructs.**

| Construct Name | VHH Name | Target | VHH Source |
|---|---|---|---|
| CsgA | N/A | N/A | N/A |
| CsgA-αGFP | NbGFP | GFP | Rothbauer *et al.*, 2006 [57] |
| CsgA-αStx2 | JGH-G1 | Shiga toxin 2 | Tremblay *et al.*, 2013 [25] |
| CsgA-αTcdA | NbTcdA | *C. difficile* toxin TcdA | Hussack *et al.*, 2011 [58] |
| CsgA-αRota | 3B2 | Rotavirus inner capsid protein VP6 | Vega *et al.*, 2013 [26] |
| CsgA-αIpaD-1 | 20ipaD | *S. flexneri* T3SS | Barta *et al.*, 2017 [20] |
| CsgA-αIpaD-2 | JPS-G3 | *S. flexneri* T3SS | Barta *et al.*, 2017 [20] |
| CsgA-αFla-3 | JUV-E8 (αFla-3) | REPEC flagellin | This study |
| CsgA-αFla-4 | JUV-G8 (αFla-4) | REPEC flagellin | This study |
| CsgA-αInt-12 | JWS-H4 (αInt-12) | EPEC intimin | This study |
| CsgA-αInt-17 | JWU-G8 (αInt-14) | EPEC intimin | This study |
| CsgA-αgp900-1 | JJ-D1 | *C. parvum* antigen gp900 | Jaskiewicz *et al.*, 2021 [59] |
| CsgA-αgp900-2 | JMP-F7 | *C. parvum* antigen gp900 | Jaskiewicz *et al.*, 2021 [59] |

[56]. Notably, cells that do not form protein fibers do not form films at all when subjected to this protocol. Previous work from our lab has confirmed the tolerance of the curli system to a wide range of CsgA fusion proteins without negatively affecting protein fiber assembly [44,56].

## CsgA-VHH-producing PBP8 binds soluble protein targets and neutralizes a bacterial toxin

Having demonstrated the formation of curli fibers on PBP8 from engineered CsgA-VHH monomers, we next tested the functionality of the fused VHH domains. To assess the ability of curli-displayed CsgA-αGFP to target GFP, the PBP8 were cultured and incubated with GFP suspended in simulated colonic fluid. After 15 minutes of incubation at 37°C, the suspensions were pelleted, and the fluorescence of the supernatant was measured to quantify the remaining unbound GFP by comparing the signal to a calibration curve of known GFP concentrations. The unbound GFP concentration was then subtracted from the initial solution concentration to estimate bound GFP, as shown in Fig 5A. No significant binding was observed in the PBP8 pellet when the unfused CsgA control or an off-target (CsgA-αRota) VHH was used. In contrast, upon pelleting PBP8 expressing CsgA-αGFP, the signal localized to the pellet was at least 56-fold higher than the negative controls, with the GFP concentration in the supernatant decreased significantly (Fig 5A).

Shiga toxins are potent bacterial toxins comprising several variants produced by the AE pathogen EHEC, as well as by serotype 1 of *Shigella dysenteriae*, and are responsible for the life-threatening manifestations of infections by these pathogens [60]. To determine if curli-VHH fibers can sequester a potent bacterial toxin from solution, we tested PBP8 producing a previously characterized VHH directed against Shiga toxin 2 (Stx2) [25]. PBP8 with curli fiber-displayed VHHs were suspended in 10 ng/mL purified Stx2 and incubated for 1 hour at 37°C. After pelleting the PBP8, monolayers of Vero cells were treated with the supernatant for 48 hours and their viability measured by a PrestoBlue cell viability assay. Supernatants derived from control PBP8 producing no CsgA, CsgA alone, or CsgA fused to either of two VHHs directed against irrelevant proteins (GFP or TcdA, an unrelated toxin) were toxic, yielding ~20% Vero cell viability. In contrast, incubation of Stx2 with CsgA-αStx2-producing PBP8 significantly reduced Stx2-induced toxicity, as ~75% of Vero cells survived treatment with supernatant (P < 0.0001; Fig 5B).

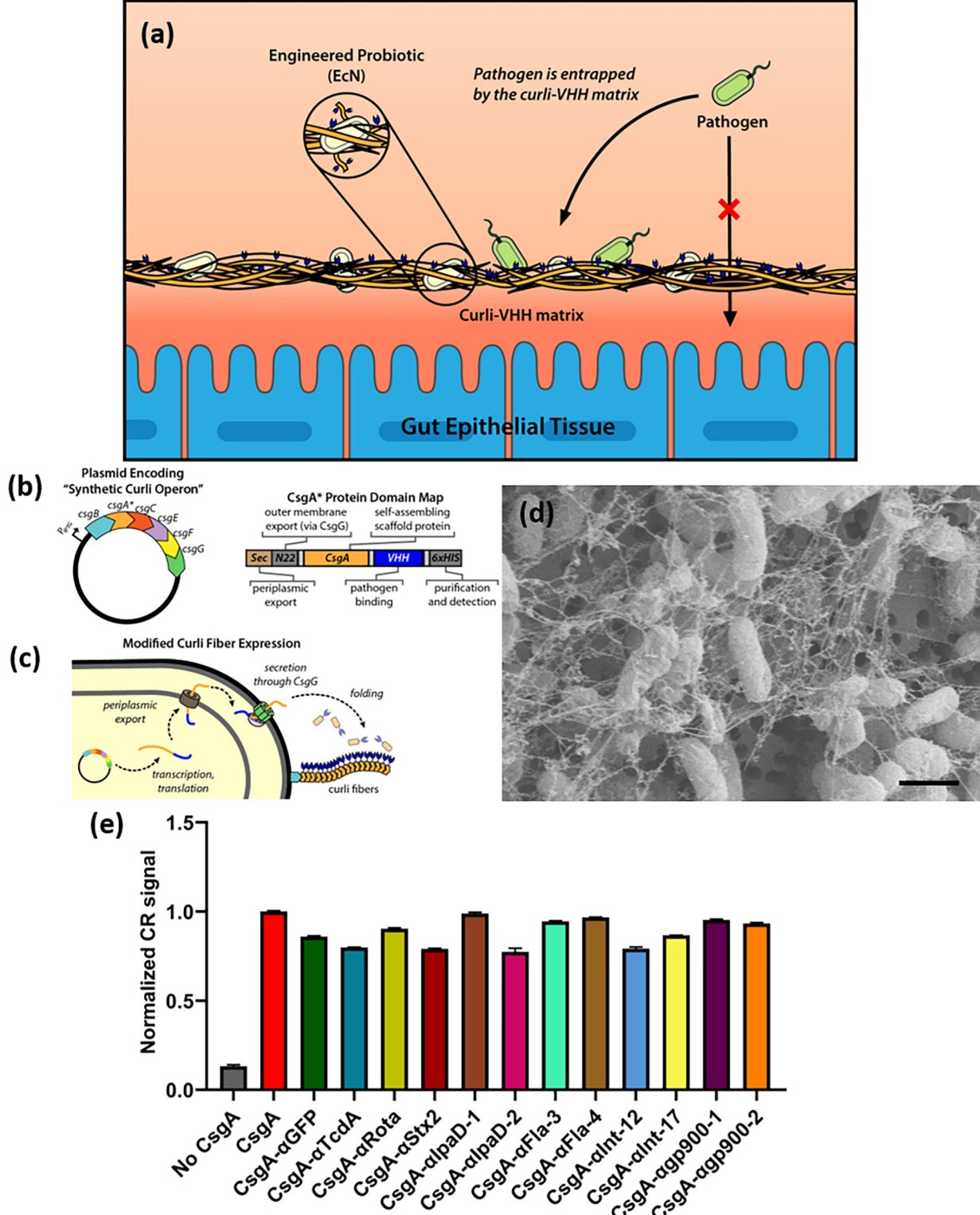

**Fig 4. Formation of VHH-functionalized curli fibers by EcN-derived strain PBP8.** (a) Plasmid map of synthetic curli operon and domain map of CsgA-VHH fusion constructs. (b) Idealized schematic of curli fiber export and self-assembly mechanism. (c) Representative scanning electron micrograph demonstrating the formation of curli fibers by CsgA-VHH-expressing PBP8 (CsgA-αgp900-2, scale bar = 1 μm). (d) Congo Red (CR) binding assays of induced cultures of PBP8 bearing plasmids encoding various CsgA-VHH constructs. CR binding is indicative of the formation of curli protein fibers. Data corresponds to $A_{490}$ measurements normalized to the CsgA positive control, and is presented as

mean ± SEM. One-way ANOVA (P < 0.0001) was performed to test presence of difference between conditions. All CsgA constructs exhibited P-values < 0.0001 compared to the "No CsgA" control, as calculated by Welch's t-test.

## CsgA-VHH-producing PBP8 binds to targets on the cell surface of bacterial enteric pathogens

To determine if PBP8 curli fiber displayed VHHs can neutralize virulence factors physically associated with pathogens, we used VHH sequences against several cell-anchored targets from multiple microorganisms and tested the ability of the VHHs to exhibit binding and/or mitigate virulence. We first generated CsgA-VHH constructs targeting the flagellar proteins of REPEC employing VHHs αFla-3 and 4 which had been shown to inhibit REPEC motility (Fig 2). To test the ability of these engineered EcN strains to bind REPEC, an aggregation assay was used. When planktonic bacterial cells are added to a conical-well plate, they gradually settle to the central point at the bottom of the conical well and can be visualized as a focused pellet. In contrast, in the presence of an aggregant, cells instead form a lattice that blankets the bottom of the well uniformly, observed as diffusely distributed cells indicating microbial aggregation. Because the efficiency of aggregation is subject to the ratio of aggregant to cells, we mixed suspensions of REPEC and PBP8 transformants at different concentrations, then photographed the plates after allowing them to settle for a day (S4 Fig). Whereas no aggregation occurred with PBP8 expressing three different control VHHs, VHHs αFla-3 and 4 caused aggregation in a concentration and VHH-dependent manner, thus demonstrating specific binding of the anti-Fla curli-VHHs to their targets. Specifically, PBP8 expressing CsgA-αFla-3 or -4 triggered more aggregation as their concentrations increased from 0.1x their overnight culture density to 0.3x, 1x, and 3x. These same PBP8 suspensions did not cause aggregation when incubated with rabbit enterohemorrhagic *E. coli* (REHEC), a strain related to REPEC but with an antigenically distinct Fla (S4 Fig), demonstrating VHH target specificity.

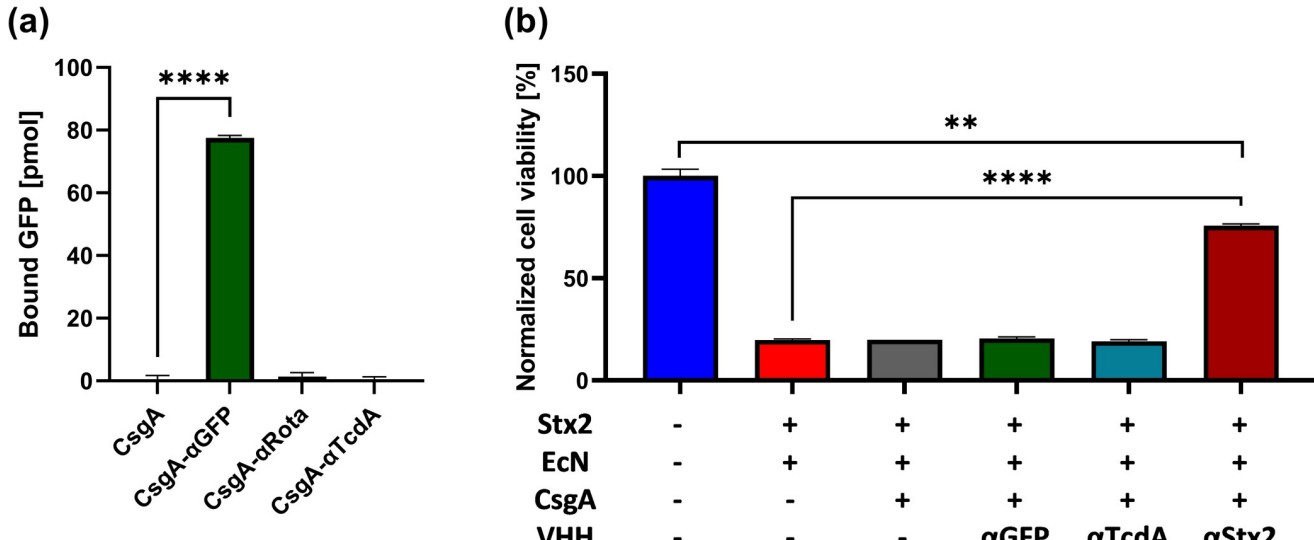

**Fig 5. CsgA-VHHs can bind soluble proteins.** (a) GFP pull-down assay. After incubation with 150 nM GFP, PBP8 CsgA-αGFP bound specifically to its target, as demonstrated by the depletion GFP in the cell supernatant. (b) Stx2 pull-down assay. PBP8 CsgA-αStx2 was used to selectively remove Stx2 upon incubation in a 10 ng/mL Stx2 solution. Supernatants were added to Vero cell monolayers and cell viability was measured. Data presented as mean ± SEM. One-way ANOVA (P < 0.0001) was performed to test presence of difference between conditions, P-values calculated by Welch's t-test. ** P < 0.01; **** P < 0.0001.

Flagella, which are flexible and extend 5–20 μm from the surface of *E. coli*, should be more easily bound by curli-displayed VHHs than structures closely associated with the microbial surface, where binding may be sterically constrained [61]. We next tested PBP8 expressing VHHs targeting Type III secretion systems (T3SSs), which extend less than 50 nm from the outer membrane of Gram-negative bacteria [62], for binding and pathogen neutralization.

*Shigella flexneri*, a significant contributor to the worldwide diarrheal disease burden [2], encodes a T3SS that is essential for virulence. IpaD, which assembles at the distal end of the T3SS apparatus, prevents premature exposure of the effectors to the extracellular environment [63,64]. IpaD has been shown to be involved in pore formation in the host membrane by regulating IpaB and IpaC, and inhibiting IpaD function has been shown to reduce *S. flexneri*'s ability to disrupt host cells [65]. Pore formation is typically evaluated using a contact-mediated hemolysis assay that measures red blood cell lysis following exposure to *S. flexneri* [66]. Two anti-IpaD VHHs (αIpaD-1 and αIpaD-2, Table 2), previously shown to inhibit red blood cell lysis by *S. flexneri* when applied as soluble proteins [20], were expressed on PBP8 curli. Both curli-VHH constructs bound to soluble IpaD, although CsgA-αIpaD-2 exhibited a much stronger signal than CsgA-αIpaD-1 as measured by ELISA (S5 Fig).

The ability of these curli-VHHs to neutralize *S. flexneri* was next tested in a hemolysis assay. Suspensions of *S. flexneri* were incubated with PBP8 expressing CsgA-αIpaD-1, CsgA-αIpaD-2, or CsgA-αGFP (negative control). As a positive control, we prepared a soluble trimer of αIpaD VHHs (Fig 6). Following incubation of PBP8 with *Shigella*, bacterial suspensions were exposed to sheep blood, and hemolysis was measured colorimetrically. PBP8 producing CsgA-αIpaD-2 abolished much of the observed hemolysis, as did the positive control. Surprisingly, we did not obtain similar results with CsgA-αIpaD-1, despite its ability to bind soluble IpaD (Fig 6), suggesting this VHH may be sterically inhibited from binding *Shigella* TS33 when expressed on curli fibers.

## CsgA-VHH-producing PBP8 can bind to the eukaryotic pathogen *Cryptosporidium parvum*

To determine if PBP8 curli fiber displayed VHHs can bind to a protein target on the surface of a eukaryotic pathogen, we utilized the parasite *Cryptosporidium parvum*, which, along with *C. hominis*, is the major cause of cryptosporidiosis. Cryptosporidiosis is an enteric diarrheal disease, a major cause of morbidity in children in low-income countries [1,2], and is the leading cause of waterborne disease in the United States with the number of infections continuing to rise [67]. Cryptosporidiosis can also be severe in immunocompromised individuals such as those living with AIDS/HIV, where the prevalence of enteric protozoan infection was reported to be 30.6% [68]. Novel treatments for cryptosporidiosis are urgently needed, as nitazoxanide, the only available treatment, has limited efficacy and is effective in only a subset of patients [69].

Because of the difficulty of studying and propagating human pathogenic *Cryptosporidium* spp., relatively few documented virulence factors or well-characterized surface-exposed antigens are known compared to many other enteric pathogens. One such antigen, glycoprotein gp900, is expressed on the surface of *C. parvum* sporozoites and has been reported to be a virulence factor due to its roles in motility and host cell invasion [70]. Importantly, gp900 is also shed in trails by sporozoites as they move about.

Previous studies identified VHHs that recognize the carboxyl-terminal domain of gp900 (αgp900-1 and αgp900-2, Table 2) [59]. We used both VHH sequences to generate CsgA fusions, produced these in PBP8, and tested their ability to bind their target antigen. We first tested gp900 binding using an ELISA, where CsgA-VHH-expressing PBP8 were adsorbed

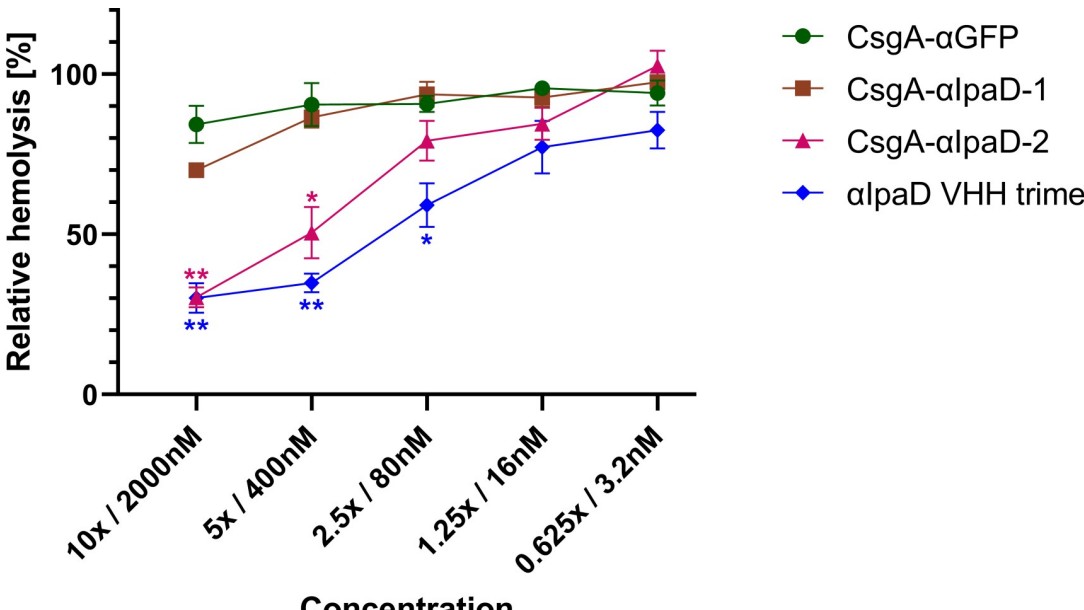

**Fig 6. CsgA-VHHs can neutralize *S. flexneri* contact-mediated hemolysis.** PBP8 expressing CsgA-αIpaD-2, but not CsgA-αIpaD-1, inhibited hemolysis of sheep red blood cells, significantly outperforming the off-target negative control (CsgA-αGFP). Data presented as mean ± SEM. Two-way ANOVA (P < 0.0001) was performed to test the presence of difference between conditions, P-values calculated by Welch's t-test. * P < 0.05; ** P < 0.01.

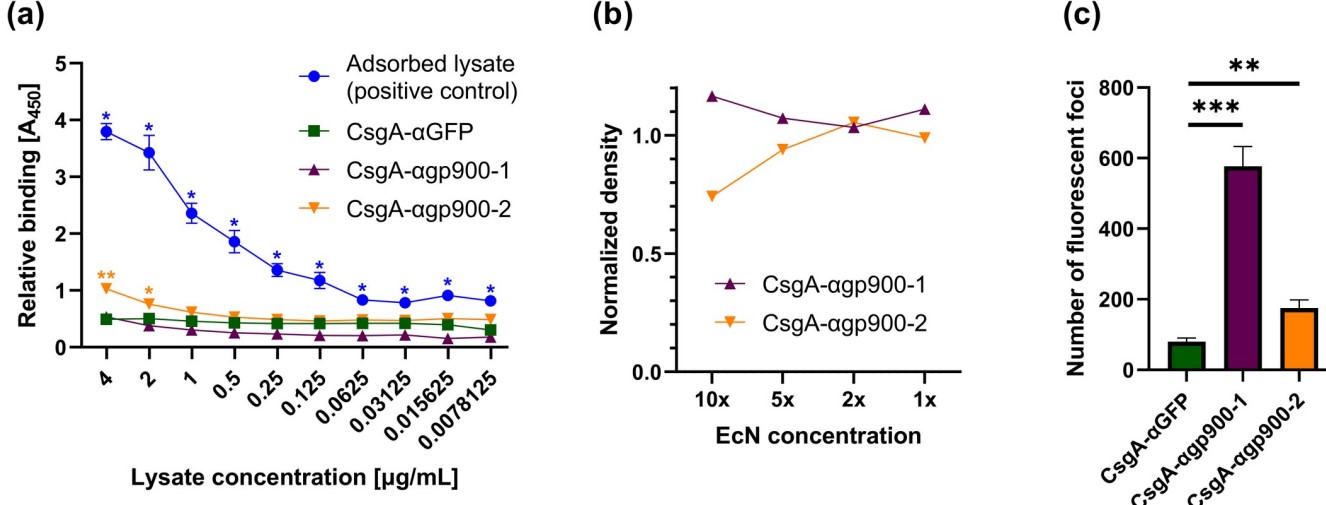

**Fig 7. CsgA-VHHs can bind to the eukaryotic pathogen *Cryptosporidium parvum*.** (a) ELISA demonstrating the ability of CsgA-αgp900-2 to bind gp900. PBP8 was adsorbed onto a well plate, followed by incubation with gp900-containing *C. parvum* lysate. In the positive control, no PBP8 was used, and the *C. parvum* lysate was allowed to directly adsorb onto the surface of the wells. Gp900, either bound to PBP8 or adsorbed to the surface, was detected using a non-competing anti-gp900 VHH, followed by an anti-Etag IgG-HRP conjugate. (b) Western blot analysis of gp900 binding to CsgA-αgp900-2. *C. parvum* lysate was incubated with PBP8 at different concentrations. After pelleting the PBP8, supernatant form was run on gel and subjected to Western blot to assess gp900 depletion. The gp900 band was detected by a specific non-competing VHH, followed by an anti-Etag IgG-HRP conjugate. Band intensity was normalized to a CsgA-αGFP negative control. (c) CsgA-αgp900-1 and -2 bind to *C. parvum* sporozoites. PBP8 was applied to immobilized and fixed sporozoites, followed by staining with anti-LPS Mab and anti-mouse IgG Alexa Fluor 568. Slides were inspected under TRITC filter and 5 images were taken under 200x magnification for each condition. Foci of PBP8 accumulation were quantified using ImageJ particle analyzer. Data presented as mean ± SEM. Two-way ANOVA (P < 0.0001) was performed to test presence of difference between conditions, P-values calculated by Welch's t-test. * P < 0.05; ** P < 0.01; *** P < 0.001.

onto plastic and incubated with *C. parvum* lysate. PBP8 that expressed CsgA-αgp900-2, though not CsgA-αgp900-1, produced a weak binding signal indicating binding at high concentrations, significantly different from the CsgA-αGFP negative control (Fig 7A). Similarly, in a pull-down assay, PBP8 expressing CsgA-αgp900-2, but not αgp900-1, depleted gp900 from *C. parvum* lysate (Fig 7B).

Binding to the parasites was then assayed by fixing *C. parvum* sporozoites onto slides, exposing them to PBP8, and quantifying the parasite-bound PBP8. Interestingly, while CsgA-αgp900-1 did not appear to bind the antigen in its soluble form, it significantly outperformed CsgA-αgp900-2 in pathogen binding, although CsgA-αgp900-2 also bound the parasites significantly better than the CsgA-αGFP negative control (Fig 7C). Qualitatively, colocalization of PBP8 CsgA-αgp900 with the fixed sporozoites was also observed (S6 Fig).

## Discussion

In this report we propose a novel approach, which we term 'curli-based pathogen sequestration', to treat or prevent enteric infectious disease. By appending VHH domains to curli fibers and displaying these modified fibers on the *E. coli* strain PBP8 (derived from the probiotic EcN), we demonstrated binding, and in some cases, neutralization, of virulence factors produced by several different pathogens. These factors include the potent enteric bacterial toxin Stx2 (Fig 5), the flagella of REPEC (S4 Fig), the T3SS of *S. flexneri* (Fig 6), and the surface antigen gp900 of the eukaryotic parasite *C. parvum* (Fig 7). In addition, we described the identification and characterization of VHHs that bind to various surface-exposed EPEC or EHEC virulence factors. These VHHs include those that inhibited flagella-driven motility (Fig 2) or interfered with Tir-intimin binding and exhibited neutralizing activity against their target pathogen (Fig 3). Adding to the growing body of validated VHH sequences, these antibodies may find use in myriad therapeutic, diagnostic, and research applications, and contribute to the study and treatment of pathogenic *E. coli*.

We propose here the further exploration of EcN curli fiber-displayed VHHs as a potential strategy for the prevention of enteric pathogen establishment. This work would benefit from further structural characterization of CsgA-VHH fibers to understand their morphology and confirm their functional production in vivo. We suggest that by maintaining a level of EcN-displayed anti-pathogen VHHs on curli fibers, the probiotic bacteria may sequester incoming pathogens in the GI tract before they reach their target sites of infection, allowing their elimination from the body by natural processes without causing pathology. The use of EcN-displayed VHHs that bind essential virulence factors, and thus neutralize the ability of the pathogen to infect the host, could add additional protection.

The efficacy of curli-based sequestration has not yet been tested *in vivo*, and requires further validation. Specifically, such validation will necessitate the development of animal infection models that support coadministration and maintenance of pathogens and EcN at stable levels, allowing for testing of the efficacy of the curli-based sequestration matrix. Nevertheless, our approach has several features that make it particularly well-suited for binding enteric pathogens *in situ*. First, the curli matrix can provide multivalent display of pathogen binding domains, as each of the stacked CsgA monomers is linked to a VHH domain. Multivalency has been shown to be an important factor for enhancing the performance of several pathogen binding systems [25,71–74]. Secondly, curli-based materials have been demonstrated to work in the gastrointestinal tract, both in the form of anti-inflammatory peptides displayed on curli fibers produced by EcN [43], as well as similarly functionalized curli hydrogels [75]. In a recent publication, we have also demonstrated expression of a CsgA-VHH fusion (CsgA-αGFP) in the gastrointestinal tract of mice [76].

Additionally, because EcN can replicate in the host and continuously produce new curli fibers, we anticipate that steady-state levels of the displayed VHH multimers can be maintained for significant periods of time despite the harsh proteolytic environment and constant flow in the gut. This *in situ* production could also have the benefit of a high local concentration near the site of pathogen colonization. Notably, the ability of EcN to maintain a steady-state density in the mammalian GI tract with regular dosing is supported by data in mice and humans [35,40]. Progress has also been made using EcN in clinical trials as an engineerable chassis organism for therapeutic applications in the gut [77–80].

Lastly, the use of engineered microbes to deliver and produce the sequestrant *in situ* may offer additional benefits. Unlike inert biomaterials, bacteria can sense their environment and respond to changes within it. As such, engineered living therapeutics can exert additional therapeutic or diagnostic functions in tandem with the production of pathogen-binding molecules–either in the form of genetic circuits or through their inherent native properties. In particular, the use of EcN as a chassis organism may prove advantageous in targeting EPEC and EHEC variants, as it has been shown to outcompete these strains in biofilm formation [38], prevent EHEC colonization in mice by occupying a similar nutritional niche [81], as well as promote intestinal health through several other mechanisms [82]. Therefore, while further studies are needed, we hypothesize that the pathogen sequestration strategy may work synergistically with EcN's probiotic functions, potentially resulting in a more effective treatment than either wild-type EcN or a non-living curli-VHH material.

Notably, not all CsgA-VHH constructs proved equally efficacious. For instance, while both anti-IpaD VHHs exhibited neutralizing activity in their soluble form [20], only CsgA-αIpaD-2 retained this ability when displayed on EcN-bound curli fibers. This difference may be due to steric effects related to the spatial arrangement of neighboring VHHs fused to CsgA domains. Alternatively, the variation may be related to CsgA-VHH secretion and curli fiber assembly as translocation of some VHH domains through the curli secretion machinery may interfere with proper protein folding. The curli secretion and assembly pathway involves multiple steps and requires translocation through the inner membrane into the periplasm via the SecYEG translocon, followed by transit through the dedicated CsgG outer membrane pore [83]. While this pathway has been shown to accommodate various CsgA fusions, the complete scope of specific limitations of this capacity are yet to be fully elucidated [84]. Nevertheless, the impressive amenability of the CsgG outer membrane pore to a wide variety of VHH fusions demonstrates the modularity and flexibility of functionalized curli in general, and VHH-based applications specifically.

In recent years, other pathogen sequestration approaches have been explored for the binding and removal of viruses from patients or the environment. Dey and coworkers developed synthetic polymer nanogels that were able to bind herpes simplex virus 1 and block its ability to infect mammalian cells *in vitro* [74]. More recently, Pu and coworkers fused an influenza-binding peptide to curli fibers and demonstrated their ability to remove virus particles from contaminated water [85]. Strategies based on feeding VHHs or VHH multimers have also been reported and have shown promise for treating the pathology of enteric pathogen infections [18,19,22].

Successful deployment of VHH-based therapeutics against enteric pathogens to locations where they are most needed will require cheap manufacturing and the need for only a limited medical infrastructure to support their effective use. Our engineered probiotic approach has the potential to satisfy these requirements. Though the bacteria may need proper formulation, they should not require substantial purification steps or an onerous cold chain for distribution. Furthermore, recent animal and human trial results suggest that daily oral administration can maintain a steady-state concentration of EcN in the GI tract [35,40], and that bacterial counts

drop rapidly after ceasing administration–a useful safety feature of this approach. The prospect of local delivery inside the gut would also minimize off-target effects and side effects that result from systemic delivery while offering the additional health benefits of conventional probiotics. Since our approach is not bactericidal, it also has interesting implications for circumventing the development of antibiotic resistance that will warrant further investigation.

## Materials and methods

### Ethics statements

All animal experiments were approved by the Tufts University Institutional Animal Care Use Committee in accordance with the Guide for the Care and Use of Laboratory Animals of the National Research Council. *Cryptosporidium* parasites used in this study were generated in animals in compliance with protocols No. G2017-107 and No. G2017-120. The VHH-display phage library was generated and derived from an alpaca in accordance with the protocol No. G2019-142.

### Cell strains and plasmids

All strains and plasmids used in this study are summarized in S1 Table.

### Bacterial culture

All *E. coli* and *C. rodentium* strains were cultured in LB broth at 37˚C at 225 RPM, unless otherwise specified. EcN (PBP8) strains were streaked from frozen stock onto selective lysogeny broth (LB) agar plates and grown overnight at 37˚C. Cultures were subsequently started from single colonies into 5 mL LB supplemented with 50 μg/mL kanamycin and grown overnight at 37˚C with shaking at 225 RPM. The following day, overnight cultures were diluted 1:100 into 10 mL fresh media and grown at 37˚C and 225 RPM, and protein expression was induced by adding 100 μM isopropyl β-D-1-thiogalactopyranoside (IPTG). Induced cultures were grown overnight.

*Shigella flexneri* was streaked from frozen stock onto tryptic soy agar (TSA) plates supplemented with 0.02% Congo Red (CR) and grown overnight at 37˚C. The following day, 3 colonies were used to inoculate 50 mL of tryptic soy broth (TSB) in a baffled flask. Only colonies stained red by CR were used. The culture was grown at 37˚C and 225 RPM to OD at 600 nm of 0.8–1.0, placed on ice upon reaching the desired OD, pelleted at 3500 RPM for 10 minutes at 4˚C, and resuspended in 5 mL to obtain a 10x suspension.

### Mammalian cell culture

HeLa cells (ATCC CCL-2) were maintained in Dulbecco's modified Eagle's medium (DMEM) with 10% fetal bovine serum (FBS) and 1% Penicillin-Streptomycin antibiotics in a 5% $CO_2$ incubator at 37˚C. For infection, 30,000 cells were seeded on 24-well plates in a volume of 0.5 mL/well. The next day, cells were gently washed with PBS before inoculating with bacteria. Vero cells (ATCC CCL-81) were grown in Eagle's Minimum Essential Medium (EMEM) supplemented with 10% FBS in a 5% $CO_2$ incubator at 37˚C. For Stx2 toxicity assays, Vero cells were seeded on 96-well plates one day prior to incubation with toxin.

### Parasite propagation

*C. parvum* oocysts, MD isolate originating from deer and passaged repeatedly in sheep and mice [86], were generated at Tufts University by propagation in CD-1 mice as described elsewhere [87], in compliance with study protocol No. G2017-107 approved by the Tufts

University Institutional Animal Care Use Committee. Prior to excystation, oocysts were bleached on ice for 7 minutes using 5% dilution of commercial bleach (Clorox Original, The Clorox Company, CA). To remove bleach, oocysts were washed three times by suspension in PBS and centrifugation ($18,000 \times g$, 2 min).

## Purification of flagella

Flagella from REPEC (E22), EPEC (E2348/69), and E10 (O119:H6) were isolated as described previously [88], with slight modification. Briefly, a single colony was transferred into 5 mL LB broth and incubated overnight at 37˚C with continuous shaking. The next day, the culture was diluted 1:100 into LB broth and grown at 37˚C to $OD_{600}$ of 0.5. 100 μL of the culture was plated onto the surface of eighty 100mm diameter LB agar plates and incubated for 24h at 37˚C. 500 μL of PBS was then added to each plate and a glass slide was used to gently scrape bacteria from the agar plate. Bacteria were collected in a centrifuge bottle. To shear flagella from the bacteria, the centrifuge bottle was manually shaken for 2 min and then shaken for 5 mins at 4˚C at 220 RPM. The bottle was then centrifuged at $7025 \times g$ for 20 min at 4˚C to remove cell debris. Bacteria-free supernatant was transferred to a new centrifuge bottle, which was further centrifuged at $25,402 \times g$ for 1 hour at 4˚C to precipitate flagella. To recover flagella, the supernatant was removed, and the pellet was resuspended in 500 μL of ice-cold PBS. To confirm that the purified flagella encompassed flagellin monomers of 60 kDa, flagella were visualized by sodium dodecyl sulphate polyacrylamide gel electrophoresis (SDS-PAGE) and by Western blotting using Rabbit anti-H6 flagella antibody. Note: EPEC E2348/69 produces fewer flagella filaments when grown in LB media [52]. Therefore, to maximize shearing of flagella from E2348/69, bacteria were either passed through a syringe and a 22-gauge needle or heat treated at 65˚C for 30 mins.

## Alpaca immunizations

Immunizations were performed essentially as described by Vrentas *et al*. [89]. Two different pairs of alpacas were each immunized in two separate rounds of immunization with various combinations of purified REPEC or EPEC flagella, and/or recombinant proteins MBP/EHEC intimin, or 6xHis/EHEC Tir. For each round of immunization, five successive multi-site subcutaneous injections were employed at about 3-week intervals. Blood was obtained for lymphocyte preparation 3–5 days after the fifth immunization and RNA was prepared from lymphocytes using the RNeasy kit (Qiagen, Valencia, CA). A VHH-display phage library was prepared essentially as described previously [73] following each of the rounds of alpaca immunization, yielding libraries with complexities of about $1–2 \times 10^7$ independent clones, and >95% containing VHH inserts.

## Identification and purification of VHHs

Phage library panning methods have been previously described [90]. Typically, the virulence factor proteins were coated onto plastic at 10 μg/mL of target in the first panning round, followed by a second round of panning at high stringency, with virulence factor proteins coated at 1 μg/mL, and using a 10-fold lower titer of input phage, shorter binding times, and longer washes. In some cases, the virulence factor targets were captured onto plastic by previously coated VHHs which bind the target or its fusion partner. VHH capture panning was also used in some cases to block isolation of VHHs to immunodominant epitopes on monomeric targets. Random clones from the selected populations were then screened by ELISA for expression of VHHs that bound to the virulence factor targets. Clones producing the strongest signals or showing broader target specificity were characterized by DNA fingerprinting. The

coding sequences of VHHs selected as having unique fingerprints and the strongest ELISA signals were obtained. Based on sequence homology, one VHH representing each homology group (having no evidence of a common B cell clonal origin) was selected for expression and characterization. These VHHs were expressed individually in pET32 vectors and purified as recombinant *E. coli* thioredoxin fusions with a carboxy-terminal E-tag, as previously described [90].

## Dilution ELISA

ELISAs were performed using Nunc Maxisorp 96 well plates (Thermo Fisher Scientific). Virulence factor targets were typically coated overnight at 4˚C, 1 µg/mL in PBS, then blocked for at least an hour at 37˚C with 4% milk in PBS, 0.1% Tween. For capture ELISAs, plates were first coated with 5 µg/mL of VHHs that recognized the virulence factor or its fusion partner. The captured VHHs lacked both the thioredoxin partner and E-tag. After blocking, the virulence factor was then incubated at 1 µg/mL for one hour at 37˚C with 4% milk in PBS, 0.1% Tween and washed. Dilution ELISAs were then initiated by diluting the VHH (expressed in a pET-32 vector with an amino terminal thioredoxin and a carboxyl terminal E-tag) to 125 nM and performing serial dilutions of 1:5. After incubation for one hour at 37˚C, plates were washed and then incubated with 1:10,000 rabbit HRP/anti-E-tag (Bethyl Laboratories) for one hour, washed, developed with TMB (Sigma Aldrich) as recommended by the manufacturer and measured for absorbance at 450 nm.

## ELISA measuring the effect of anti-Tir VHHs on the intimin-Tir interaction

The ability of anti-Tir VHHs derived from EHEC to interfere with intimin-Tir binding was measured by ELISA. High-binding assay plates (Corning) were coated with 5 µg/mL of recombinant his-tagged Tir diluted in 1x coating buffer (50 mM $Na_2CO_3$, 50 mM $NaHCO_3$, pH 9.6) in a volume of 100 µL per well and incubated overnight at 4˚C. Plates were washed three times with 300 µL of wash buffer (0.05% Tween in PBS) and then blocked with BSA (3% in PBS) for 2 hours at room temperature (RT). Plates were washed and 100 µL of 500 nM anti-Tir VHH were added to each well. 0.1% BSA was used as a negative control. Plates were incubated at RT for 2 hours or at 4˚C overnight. Wells were then probed with 150 nM GST-tagged intimin or with GST alone, and incubated at RT for 2 hours or at 4˚C overnight. Plates were washed again and then fixed with 3.7% paraformaldehyde at RT for 20 mins at 4˚C. Following another wash, plates were blocked with 5% milk in PBS for 30 min at RT. After washing, plates were incubated with goat anti-GST (GE Healthcare) for an hour and GST binding was detected kinetically using an alkaline-phosphatase-linked rabbit anti-goat IgG secondary antibody (diluted 1:2000 in 0.1% BSA/PBS). Binding of the secondary antibody was detected colorimetrically (AP substrate N1891, Sigma Aldrich) at 405 nm, and the average reaction rate ($V_{mean}$) was calculated.

## REPEC motility assay

Motility assays were performed as described previously [73], with slight modification, to measure the ability of anti-Fla VHHs to inhibit Rabbit Enteropathogenic *Escherichia coli* (REPEC) motility. Briefly, REPEC cultures were streaked on LB agar plates and incubated for 16 hours at 37˚C. The next day, a single colony was transferred into 5 mL LB broth and incubated overnight at 37˚C with continuous shaking. On the following day, the culture was diluted 1:50 into LB broth and grown at 37˚C with continuous shaking to an $OD_{600}$ of 0.5. A 1:1 dilution of bacteria and VHHs was then prepared (6 µL of bacterial culture was mixed with 6 µL of VHH

concentrations ranging from 0 to 6.4 μM), mixed gently with a pipette, and incubated at 4˚C for 2 h. The 12 μL mixture was then transferred to the center surface of a 0.3% semi solid agar plate and incubated for 24 h at room temperature. The diameter of bacterial growth was measured by first placing the plate on a dark background to enhance the contrast between bacterial growth and the agar medium. A metric scale ruler was then used to measure the growth diameter. Images were captured using a Syngene imager.

### EPEC pedestal assay

The ability of anti-intimin and anti-Tir VHHs to inhibit EPEC pedestal formation was assessed after infection of HeLa cells, as described previously [91], with slight modification. Briefly, 30,000 HeLa cells were inoculated into the wells of 24-well plates (Invitro Scientific,) and incubated overnight at 37˚C in an incubator with 5% CO2. On the same day, a single EPEC colony was inoculated into 5 mL DMEM in 100mM HEPES medium (pH 7.4) and incubated overnight in 5% CO2 at 37˚C without shaking. The next day, the EPEC culture was diluted 1:16 into new infection medium (0.6 μL EPEC added to 9.4 μL media containing DMEM, 20 mM HEPES, and 3.5% FBS; pH 7.4), and 3.33 μL of the EPEC suspension were incubated either alone or with 100 nM anti-Tir or anti-intimin VHH in 0.5 mL DMEM at 4˚C for 2 hours, in 1.5 mL Eppendorf tubes on a rocker. HeLa cell monolayers were then washed with 0.5 mL PBS, and EPEC suspensions were added to the monolayers. Plates were then centrifuged at 500 RPM for 5 min and incubated for 3 h at 37˚C in a 5% CO2 incubator. Next, cells were washed twice with PBS, fixed with 0.5 mL 2.5% paraformaldehyde in PBS for 10 mins at RT on a shaker, washed twice with PBS for 5 min on a shaker at RT, permeabilized with 0.5 mL of 0.1% TritonX-100 for 5 min, and washed twice again before staining with DAPI (Thermo Fisher Scientific) and Alexa Fluor-488 Phalloidin (Thermo Fisher Scientific) at RT for 1.5 hours. Monolayers were then washed and 7 μL prolong gold anti-fade reagent (Thermo Fisher Scientific) was used to mount coverslips on wells before imaging with a fluorescent microscope.

EPEC pedestal formation was blindly scored, as follows; 1: Very few pedestals are present on the edges of the wells, 2: More pedestals present, only at the edges of the well, 3: Most cells have no pedestals, but a few pedestals present in the center and edges of wells, 4: Most cells have pedestals, but a few empty cells are present, 5: The majority of the cells have pedestals. Using the above numbering criteria, pedestals were scored blindly by a second researcher.

### CsgA-VHH plasmid construction and cloning

The cloning of the synthetic curli operon *csgBACEFG* onto the pL6FO vector was described in detail elsewhere [55]. DNA sequences of desired VHHs and corresponding primers were synthesized by and purchased from Integrated DNA Technologies. Plasmid construction was carried out using Gibson Assembly [92].

### Quantitative Congo Red binding assay

Curli fiber formation was quantified using a Congo Red binding assay as previously described [43]. Briefly, 1 mL of induced EcN CsgA-VHH culture was pelleted at 4000 × g for 10 minutes at room temperature and resuspended in a 25 μM Congo Red PBS solution. After a 10-minute incubation, the cell suspension was pelleted again, and the unbound Congo Red dye was quantified by measuring the supernatant absorbance at 490 nm. The signal was subtracted from a Congo Red blank, divided by the culture's $OD_{600}$ measurement to reflect curli production per cell, and normalized with respect to a EcN CsgA (no VHH) positive control.

## Electron microscopy

Field emission scanning electron microscope (FESEM) samples were prepared by fixing with 2% (w/v) glutaraldehyde and 2% (w/v) paraformaldehyde at room temperature, overnight. The samples were gently washed with water, and the solvent was gradually exchanged with ethanol with an increasing ethanol 15-minute incubation step gradient (25, 50, 75 and 100% (v/v) ethanol). The samples were dried in a critical point dryer, placed onto SEM sample holders using silver adhesive (Electron Microscopy Sciences) and sputtered until they were coated in a 10–20 nm layer of Pt/Pd. Images were acquired using a Zeiss Ultra55 FESEM equipped with a field emission gun operating at 5–10 kV.

## GFP pull-down assay

For each condition, 1 mL of induced overnight culture was centrifuged at $4000 \times g$ for 10 minutes. The supernatant was aspirated, and the pellets were resuspended in 150 nM GFP in fasted-state simulated colonic fluid, which was prepared as described by Vertzoni *et al*. [93]. The cells were incubated on a shaking platform (225 RPM) at 37˚C for 15 minutes and pelleted again. The GFP remaining in solution was assayed by measuring the fluorescent signal (485nm/528nm) using a plate reader (Spectramax M5, Molecular Devices). GFP concentration was estimated based on a calibration curve using known GFP concentrations.

## Shiga toxin pull-down assay

Induced EcN cultures were pelleted at $4000 \times g$ for 10 minutes and resuspended in 10 ng/mL of Stx2 in PBS. The bacterial suspensions were then serially diluted tenfold (from 1:10 to $1:10^4$) in 10 ng/mL Stx2 PBS solution, maintaining a constant Stx2 concentration. The suspensions were incubated at 37˚C on a 225 RPM rotating platform for 1 hour and pelleted again. For each condition, 10 μL of supernatant were added to 90 μL of Vero cell medium in its corresponding well. After a 48-hour incubation at 37˚C with 5% $CO_2$, 10 μL of PrestoBlue Cell Viability Reagent (Thermo Fisher Scientific) was added into each well, followed by a 10-minute incubation and measurement of fluorescent signal at 560nm/590nm.

## REPEC and REHEC aggregation assays

REPEC, REHEC and PBP8 were cultured as previously described. Cultures were pelleted and resuspended in PBS to obtain 3x, 1x, 0.3x or 0.1x suspensions as compared to the original culture density. Cell suspensions were subsequently mixed and added into 96-well conical-bottom microwell plates (Thermo Fisher Scientific) and allowed to settle overnight at room temperature prior to imaging.

## Generation of anti-IpaD VHH trimer

VHH heterotrimer was designed and generated as previously described [94,95]. Briefly, DNA encoding the 20ipaD, JMJ-F5, and JPS-G3 VHHs [20] separated by 15-amino acid flexible glycine-serine linkers ((GGGGS)₃) was synthesized (GenScript Biotech, Piscataway, NJ) and ligated into pET32b(+) vector between the N-terminal thioredoxin (trx) fusion partner and a C-terminal E-tag epitope. VHH trimer was expressed in *E. coli* Rosetta-gami 2(DE3)pLac1 (Novagen) by overnight incubation in 1 mM IPTG, followed by lysis and purification on nickel agarose resin (Invitrogen). Bound VHHs were eluted from resin using increasing concentrations of imidazole ranging from 10mM to 250mM.

## Shigella contact-mediated hemolysis assay

To determine the ability of CsgA-VHH to inhibit *Shigella* virulence activity, a contact-mediated hemolysis assay was carried out as previously described [65], with slight modification. Prior to exposure of *S. flexneri* to red blood cells, induced PBP8 cultures were pelleted and resuspended in PBS to obtain a 10x concentrated cell suspension, which was subsequently serially diluted to yield 5x, 2.5x, 1.25x and 0.625x suspensions. The pathogen was then incubated for 30 min at room temperature with either the PBP8 suspensions or the abovementioned anti-IpaD VHH heterotrimer (trx/20ipaD/JMJ-F5/JPS-G3/E) as a positive control, in concentrations between 3.2–2000 nM.

## Preparation of *Cryptosporidium* lysate

Pre-bleached *C. parvum* oocysts were excysted in 0.75% taurocholic acid suspension in PBS for 1h at 37˚C. Following centrifugation ($18,000 \times g$, 2 min), supernatant was collected and the pelleted sample consisting of sporozoites, unexcysted oocysts and oocyst shells was then sonicated (Qsonica CL5, Qsonica Sonicators, USA) with thirty cycles, 20 seconds each. Sonicated pellet was resuspended in supernatant and saved as '*C. parvum* whole lysate'. The concentration of the antigen fractions was determined by measurement of optical density using a Nanodrop instrument (ND-1000, NanoDrop Technologies).

## Pull-down of *C. parvum* antigens using *E. coli* Nissle

A variety of modified pull-down studies utilizing the principles of ELISA, Western blot and immunofluorescence were applied to test the ability of anti-gp900 VHHs fused to PBP8 curli to bind their targets. All experiments used a nonspecific control PBP8 construct which expressed curli in fusion with a VHH targeting the green fluorescent protein (CsgA-αGFP).

For the ELISA, the goal was to pull-down *C. parvum* antigens by PBP8 immobilized to a plastic surface. Briefly, 100 µl of induced overnight PBP8 cultures expressing CsgA-αgp900-1 and -2 were coated on 96-well MaxiSorb plates at 2x concentration and incubated overnight at 4˚C. The following day, plates were washed with TBS-0.1% Tween and blocked with 4% milk-TBS-0.1% Tween solution for 1 h at 37˚C. Plates were washed and *C. parvum* whole lysate was applied in 2-fold dilutions starting with 50 µg/mL concentration, and then incubated for 1 h at 37˚C. After washing, specifically bound *C. parvum* antigen was incubated with a second E-tagged detection VHH that binds to the same *C. parvum* antigen recognized by the PBP8 displayed VHH, but to a non-competing epitope, at 1 µg/mL for 1 h at 37˚C. Plates were then washed and incubated with an anti-E-tag HRP antibody (Bethyl Laboratories) at 1:10,000 for 1 h at 37˚C. Plates were washed a final time and OPD was added to each well for 20 minutes. The reaction was stopped with 1 M H2SO4 and absorbance was measured at 490 nm using a microplate reader.

For the Western blot, the goal was to quantify depletion of the target in the soluble whole lysate of *C. parvum* after incubation with PBP8 displaying an anti-gp900 VHH and removal from the solution by centrifugation. For target pull down, 50 µL of the induced (for VHH display) and blocked PBP8 was suspended in PBS at 2x concentration and incubated with 30 µg of *C. parvum* whole lysate for 1 h with rotation at room temperature. Samples were centrifuged (5000 ×g, 1 min) and supernatant was collected for analysis. Fifteen µL supernatant aliquots of supernatant were diluted with 4xLDS buffer (Novagen) to achieve 1x concentration and denatured at 70˚C for 10 minutes. Samples were loaded into the wells of 4–12% Bis-Tris gel (Novex) and electrophoresed in 1x MOPS buffer at 100 V for 10 minutes and then at 200 V for 40 minutes. The gel was transferred on the nitrocellulose membrane using a wet transfer system (395 mA, 4h). Membranes were blocked with 4% milk-TBS 0.1% Tween for 1h and

washed with TBS-T before blotting with a second detection VHH (recognizing a non-competing epitope on the target) at 1 μg/ml for 1 h with rotation. Membranes were then washed and incubated with a secondary anti-E-tag HRP antibody at 1:5,000 dilutions for 1 h with rotation. Western blots were developed using chemiluminescent substrate (GE Healthcare) and imaged using a ChemiDoc system (Bio-Rad). A densitometry analysis was performed using Image Lab software to report the percent of target band depletion as normalized to the loading control.

Immunofluorescent imaging was used to quantify PBP8 bacteria attached to *C. parvum* parasite and its trails immobilized on the surface. Pre-bleached *C. parvum* oocysts were suspended in 0.75% taurocholic acid and excysted in a 37˚C water bath for 30 minutes to release sporozoites. Aliquots of excysted 10,000 oocysts were transferred onto poly-L-lysine slides (Chromaview) and incubated for another 30 minutes at 37˚C under humidified conditions to allow for further excystation and gliding of sporozoites. Slides were then dried, fixed with 4% paraformaldehyde at room temperature (20 min) and washed with PBS. Such prepared parasites were then probed with 200 μL of 2x PBP8 suspensions and incubated for 1h at room temperature, after which they were washed with PBS to remove unbound PBP8. To detect PBP8 bound to the sporozoites and trails, slides were probed with anti-LPS Mab (ThermoFisher Scientific) at 1:200 dilution, followed by an anti-mouse IgG Alexa Fluor 568 antibody (Invitrogen) at 1:500 dilution, both incubated for 1h at room temperature. Sporozoites were counterstained with an E-tagged VHH targeting gp900 at the apical complex and trails (CsgA-αgp900-2) at 1 μg/mL concentration, followed by an anti-E-tag-FITC antibody (Bethyl Laboratories) at 1:100 dilution, both incubated for 1h at room temperature. Lastly, slides were washed, dried, and mounted with antifade medium. Fluorescing sporozoites were imaged under epifluorescence (Nikon Eclipse Ti-E microscope, Nikon Instruments Inc.). The number of fluorescent foci was quantified using ImageJ 1.48v particle analyzer (U.S. National Institutes of Health, Bethesda, Maryland, USA).

## Statistical analysis

All statistical analyses were performed using Prism 9.1.0 (GraphPad Software). Data are presented as mean ± standard error of mean (SEM), unless otherwise specified. Statistical significance was assessed using one-way or two-way analysis of variance (ANOVA), followed by Welsh's t-test, as described in figure legends.

## Supporting information

**S1 Table. Strains and plasmids.**
(DOCX)

**S2 Table. Aligned sequences of VHHs binding pathogenic *E. coli* virulence factors.** Complementarity determining regions (CDR1, 2 and 3) are highlighted and appear in order from left to right.
(PDF)

**S1 Fig. Assessing the binding properties of selected VHHs via ELISA.** Antigens used were homologues of Fla (a-b), Tir (c-g), Int (h) or EspA (i-l) corresponding to either EPEC (b, c, h), EHEC (e, g), REPEC (a, d, i, j) or *C. rodentium* (f, k, l). In each assay, antigen was either directly added to the plate in purified form (a-b, e-g, i-l), bound to the plate by an adsorbed noncompeting VHH (c-d), or displayed on the surface of MC1061 (h).
(TIFF)

**S2 Fig. Anti-EspA VHHs did not inhibit pedestal formation.** HeLa cells were exposed to EPEC incubated with VHH, fixed and stained with DAPI (blue) and Alexa Fluor-488

Phalloidin (green). Similar to the "no VHH" negative control (a, c, e), all anti-EspA VHHs tested (b, d, f) resulted in the formation of pedestals (though not all anti-EspA VHHs were tested) (scale bar = 100 μm).
(TIFF)

**S3 Fig. Representative FESEM images of PBP8 expressing CsgA and CsgA-VHH.** (a) PBP8 with no plasmid, expressing no curli fibers. (b-h) PBP8 expressing CsgA-VHH, exhibiting a range of fiber morphologies. (b) CsgA-αStx2, (c) CsgA-αInt-12, (d) CsgA-αInt-17, (e) CsgA-αFla-3, (f) CsgA-αFla-4, (g) CsgA-αIpaD-1, (h) CsgA-αgp900-2 (scale bar = 2 μm).
(TIFF)

**S4 Fig. PBP8 expressing CsgA-αFla can induce REPEC aggregation.** Suspensions of PBP8 expressing CsgA-VHH were mixed with either REPEC (a) or REHEC (b) and allowed to settle overnight in conical 96-well plates. Aggregation was only observed when REPEC was mixed with CsgA-αFla-3 and -4.
(TIFF)

**S5 Fig. PBP8 expressing CsgA-αIpaD can bind soluble IpaD.** ELISA demonstrated the ability of CsgA-αIpaD to bind soluble IpaD. PBP8 was adsorbed onto a well plate, followed by incubation with varying IpaD concentrations. Binding of IpaD to the adsorbed PBP8 was then detected by a specific non-competing VHH (JMK-H2, Barta *et*, *al*., 2017 [20]), followed by an anti-Etag IgG-HRP conjugate. PBP8 expressing either CsgA-αIpaD-1 or CsgA-αIpaD-2 significantly outperformed the off-target negative control (CsgA-αGFP). Data presented as mean ± SD. Two-way ANOVA ($P < 0.0001$) was performed to test the presence of difference between conditions, P-values calculated by Welch's t-test. * $P < 0.05$; ** $P < 0.01$; *** $P < 0.001$; **** $P < 0.0001$.
(TIFF)

**S6 Fig. PBP8 expressing CsgA-αgp900 exhibit increased attachment to and colocalization with *C. parvum* sporozoites.** (a) Fluorescent micrographs demonstrating increased attachment of PBP8 (red) to *C. parvum* sporozoites, counterstained in green in the bottom panels (scale bar = 50 μm). (b) While PBP8 (red) expressing CsgA-αgp900-1 (and to a lesser extent CsgA-αgp900-2) consistently colocalize with sporozoites (green), the CsgA-αGFP negative control was often observed away from the green fluorescent foci, consistent with nonspecific binding (scale bar = 5 μm).
(TIFF)

**S7 Fig. Curli fiber production in growing bacterial cultures as measured by Congo Red fluorescence.** A shift in emission maximum for Congo Red upon binding to curli fibers in growing PBP8 cultures was used to help confirm fiber formation for two CsgA-VHH fusions. "No curli" refers to a negative control wherein PBP8 was transformed with a plasmid bearing the same antibiotic selection markers but no curli genes. "LB" refers to a negative control containing no cells, only LB medium and Congo Red dye.
(PDF)

**S8 Fig. WAXS of dried films composed of CsgA and CsgA-VHH fusions.** (a) CsgA-6xHis; (b) CsgA-αFla-3; (c) CsgA-αgp900-2.
(PDF)

## Author Contributions

**Conceptualization:** Ilia Gelfat, Charles B. Shoemaker, John M. Leong, Neel S. Joshi.

**Data curation:** Ilia Gelfat.

**Formal analysis:** Ilia Gelfat.

**Funding acquisition:** Charles B. Shoemaker, John M. Leong, Neel S. Joshi.

**Investigation:** Ilia Gelfat, Yousuf Aqeel, Jacqueline M. Tremblay, Justyna J. Jaskiewicz, Anishma Shrestha, James N. Lee, Shenglan Hu, Xi Qian, Loranne Magoun, Abhineet Sheoran, Daniela Bedenice, Avinash Manjula-Basavanna, Amanda R. Pulsifer, Hann X. Tu, Xiaoli Li.

**Project administration:** Ilia Gelfat, Marilyn L. Minus, Marcia S. Osburne, Charles B. Shoemaker, John M. Leong, Neel S. Joshi.

**Supervision:** Saul Tzipori, Charles B. Shoemaker, John M. Leong, Neel S. Joshi.

**Validation:** Colter Giem, Amanda R. Pulsifer.

**Visualization:** Ilia Gelfat, Colter Giem, Neel S. Joshi.

**Writing – original draft:** Ilia Gelfat, Marcia S. Osburne, Charles B. Shoemaker, John M. Leong, Neel S. Joshi.

**Writing – review & editing:** Ilia Gelfat, Justyna J. Jaskiewicz, Charles B. Shoemaker, John M. Leong, Neel S. Joshi.

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
