## [Decision Letter · Decision Letter 0]

29 Jun 2021

Dear Prof. Joshi,

Thank you very much for submitting your manuscript "Single domain antibodies against enteric pathogen virulence factors are active as curli fiber fusions on probiotic E. coli Nissle 1917" for consideration at PLOS Pathogens. As with all papers reviewed by the journal, your manuscript was reviewed by members of the editorial board and by several independent reviewers. In light of the reviews (below this email), we would like to invite the resubmission of a significantly-revised version that takes into account the reviewers' comments. In particular, reviewers felt that the paper would be substantially bolstered if the functionality of the VHH-modified curli fibers was better defined and if expression of these fibers by the recombinant Nissle strain was validated within the GI tract.

We cannot make any decision about publication until we have seen the revised manuscript and your response to the reviewers' comments. Your revised manuscript is also likely to be sent to reviewers for further evaluation.

Sincerely,

Matthew A Mulvey, Ph.D.

Associate Editor

PLOS Pathogens

Brian Coombes

Section Editor

PLOS Pathogens

Kasturi Haldar

Editor-in-Chief

PLOS Pathogens

orcid.org/0000-0001-5065-158X

Michael Malim

Editor-in-Chief

PLOS Pathogens

orcid.org/0000-0002-7699-2064

Reviewer's Responses to Questions

**Part I - Summary**

Reviewer #1: The manuscript describes the development of camelid variable heavy chain (VHH) "nanobodies" to antigens from intestinal pathogens such as E. coli, Shigella, Citrobacter, and Cryptosporidium, or to toxins such as TcdA or Stx2. They also used VHHs from other studies. A subset of the VHHs were expressed from E. coli Nissle as a fusion to curli expressed on the bacterial surface. They found that the VHHs on the surface of E coli Nissle were functional, for example, to inhibit EPEC binding or toxicity to Stx2.

This study provides a nice proof-of-concept design (though some with antibodies generated in other studies) that a probiotic could express functional antibody to pathogens or toxins on the bacterial surface. Overall the manuscript is well written. One potential issue is that there does seem to be some effect of the VHH on the morphology of the curli (S3) that may influence functionality. The FESEM suggests differences in structure of the biofilm/curli that may affect the way the VHH is able to function that was not reflected in the congo red binding or ELISA assays. For example, the anti-Cryptosporidium and one of the anti-IpaD did not function as hoped even though those antibodies worked in ELISA and the curli were expressed as shown by congo-red binding. Is there a way to try to evaluate the potential functionality by another assay such as crystal violet staining as is done for biofilm? Would it be reflective, or is the issue more with structures that are attached to the target bacteria such that this approach is best for secreted virulence factors or factors that stick out further from the bacteria such as the flagellin?

Reviewer #2: In this manuscript, Gelfat and coworkers functionalize curli fibers with VHHs (Nanobodies) that bind to various bacterial virulence factors. They describe a phage display procedure which selected for relevant VHHs. The VHHs were shown to bind to soluble virulence factors and in some cases prevent toxicity. The VHHs were then fused to CsgA to create functionalized curli fibers secreted by the commensal species E. coli Nissle. This new probiotic was characterized for its ability to bind and neutralize the effect of several virulence factors. This study details exciting preliminary work which should hopefully be followed up with in vivo experiments proving its efficacy within an animal model.

Reviewer #3: Manuscript by Gelfat et al. investigates the potential use of VHHs, the variable domains of camelid heavy-chain-only antibodies as a way to inhibit the virulence of multiple pathogenic enteric bacteria. Authors initially concentrate on EPEC and identify VHHs that are active against flagellin, intimin and TIR proteins as means of blocking bacterial virulence using in vitro assays. Once several VHHs are identified the authors utilize probiotic E. coli Nissle and fuse VHHs on curli fibers. However, at this stage of the study in addition to 3 EPEC virulence factors, they add additional virulence factors to expand their study including STx2 from Shigella, TcdA from C. difficile, ipaD from Shigella agp from C. parvum. They demonstrate utilizing various in vitro assays and identify VHHs are active and inhibitory to various virulence factors. Although the study is exciting and has potential for in vivo delivery of inhibitory VHHs to block virulence in several bacterial species, critical data is missing to confirm that this delivery platform could be used in vivo. It would be powerful, instead of showing that the system is working for blocking several pathogens, to show that for the three initial virulence factor that this system could be used in vivo. Furthermore, no mechanistic information is included on how CsgA-VHH fibers are assembled. It would be informative to show whether the VHH fibers hinder with the amyloid properties of curli or if curli can still form the amyloid fibrils. As the paper is dense, it is hard to follow the manuscript and it would help if more details on methods, strains and growth conditions are provided.

**Part II – Major Issues: Key Experiments Required for Acceptance**

Reviewer #1: There are concerns with Fig 7 & S6. The PBP8 with CsgA-VHH were overlaid onto the sporozoites. Then the fluorescent foci were counted with Image J. If however, the bacteria were clumped together much more for the gp900-1, then the foci would be larger and counted as more--is that correct? There is certainly a lot more red staining for the PBP8gp900-1--is that more binding to the sporozoite, or more bacteria binding to each other? The zoomed in image or are there just more bacteria in those fields. A control of bacteria without the sporozoites might be helpful, as well as a comparison of the FESEM for both of the PBP8 gp900 constructs.

The soluble anti-Int17 was reported to reduce actin accumulation (did not show pedestals). There was a PBP8 construct made with the anti-Int17. Why was the PBP8 CsgA-anti-Int17 not tested in an actin accumulation assay? This is achievable without much additional work and would be a really nice addition to the proof of concept.

Other concerns:

1. line 219, S2 and elsewhere--what is shown is actin accumulation, not pedestal formation--though the two are related of course.

2. Figure 4 - most of it unneeded, especially with Figure 1; all of figure 4 could go into the supplemental. Alternatively, Fig 1 could go into the supplemental, since the ultimate goal is the probiotic, not the soluble VHHs.

3. Figure 4 statistical test should be a one-way ANOVA rather than two way--the only variable is congo red binding. Also, Figure 5 -panel A - one-way ANOVA

4. line 366 - does the fact that gp900 is shed in trails mean that the antibody would be titered away from the sporozoite? It seems like an antigen that does not shed would be a better target.

Reviewer #2: Major Issues - Please detail the key new experiments or modifications of existing experiments that should be absolutely required to validate study conclusions. Generally, there should be no more than 3 such required experiments or major modifications for a "Major Revision" recommendation. If more than 3 experiments are necessary to validate the study conclusions, then you are encouraged to recommend "Reject".

Reviewer #3: 1-Authors speculate the curli would be expressed in the intestinal tract but there are no studies that showed that E. coli Nissle expresses curli. Again Fig4A is speculative and should be removed. It would strengthen the manuscript to investigate at least for one of the virulence factors that the curli-VHH display system would work in vivo rather than showing it works for all in vitro. Alternatively, more information on the structural assembly or the interactions of the curli-VHH display system with virulence factors could be presented.

2- Although SEM pictures are showing the expression of curli, it is important to provide data on whether curli can still assemble into amyloid fibrils while carrying VHH. Thioflavin T based amyloid assay supported by TEM or Circular dichoism would be powerful to show whether curli can still assemble into a fibrillar conformation. Are there any data to support the VHHs are displayed as depicted in Fig4C? This figure needs to be supported by additional data to show the mechanism of action.

**Part III – Minor Issues: Editorial and Data Presentation Modifications**

Reviewer #1: 1. line 28 - delete "scherichia" in front of period so that it reads E. coli

2. line 58 - "mucus-establishing" means?

3. line 139-intimin does vary in the C-terminal third or so--hence whey there are so many different intimin types

4. line 155 - is "species" meant here, or perhaps "pathotype"? It is confusing because the previous part of the sentence is talking only about E. coli. Or is it meant that antibody to E. coli intimin does not react with Citrobacter intimin?

5. Table 1 footnotes--in "d", "e", and "f" Figure 3 not Fig 2, correct?

6. line 190 - bacterial movement diameter?

7. line 193 - Table 1 does not give IC50 values. Is Fig 2B meant?

8. Figure 4 is referred to out of order in the text. However, see above for other comments about Figure 4.

9. line 275, 626, supplemental figs S3-5 and elsewhere--EcN is referred to as the strain used, but it really is PBP8--check throughout manuscript.

10. line 287 Not all S. dysenteriae have Shiga toxin - only serotype 1. Additionally, the antibody shown only neutralizes Stx2, not Shiga toxin.

11. line 350 'tclam"?

l2. line 474 please insert the Table #s

13. line 544 - add pET32 vector to the supplemental table

14. line 575 Italicize Escherichia coli here--also check Supplemental Table and citations for italics.

15. Figure S3-it would be nice to have a figure of PBP8 expressing CsgA with no VHH.

16. Why is simulated colonic medium used for the GFP pull down study?

Reviewer #2: · The paragraph spanning lines 94-101 describes potential issues for the application of the present work. It should either be moved to the Discussion section or omitted entirely.

· The phrase “with an IC50 as low as 1.0 μM (Table 1)” found on line 192-193 calls out to Table 1 where EC50’s are displayed. It is also a little confusing which EC50’s are being referred to.

· The section title “Specific anti-Tir and anti-intimin VHHs block the interaction between intimin and Tir” suggests that results associated with anti-intimin VHHs would be discussed. However, Fig 3a only shows data including anti-Tir VHHs. This data should be included.

· The sentence “This clustering promotes F-actin assembly beneath bacteria bound at the host plasma membrane and results in the formation of actin pedestals that facilitate pathogen colonization” on line 212-214 should have a reference.

· The VHH called αTir8 was shown to be very effective at blocking the Tir-intimin interaction in vitro (Fig 3a). Why was αTir8 not used in the in vivo fluorescence assays shown in Fig 3b? Similarly, why does αTir13 perform well in vivo and yet does not block the interaction in vitro?

· Fig 4a was never referred to.

· The paragraph on lines 353-361 is not results and should be moved to the Introduction or Discussion section.

· Fig S6a and S6b seem to be redundant, only S6b is necessary.

· It would be beneficial to reader comprehension if some figure references would be included in the Discussion section.

Reviewer #3: Regarding Table 1. This is a nice summary of the results on a big data set. However, it is important also for the reader to understand how this data is generated with appropriate example. For instance show data on how the EC50 values were determined. Not all data but selective VHHs can be showed as graphs to support the tables. The rest can be left in the supplementary data

Fig1 is not very useful as the authors investigate virulence factors of several different pathogenic bacteria in the manuscript. This figure could be moved to a model slide at the end if needed.

What is the reasoning in using curli as a display system for VHHs? Can authors add this information in Ln244.

Are there any supporting data to show that the VHHs are displayed as depicted in 4C?

Fig 4A is not referred in the text. This figure is not needed where it is. Also the authors do not show any data suggesting that curli would be expressed through the epithelial lining and trap bacteria. This is speculative. Maybe move 4C to 4A and make 4B bigger so that the cloning strategy is visible.

4E no IPTG control is missing. CR also binds to cellulose. It is important to add the controls to the assay.

4D. show also no IPTG control (or CsgA) in the SEM picture to show that there is no CsgA expression.

Ln 425. Add reference to the statement that ECN expresses curli in the intestinal tract. Has this been shown?

Add details for panning into the material methods for the generation of VHHs.

PLOS authors have the option to publish the peer review history of their article (what does this mean?). If published, this will include your full peer review and any attached files.

Reviewer #1: No

Reviewer #2: No

Reviewer #3: No
---

## [Decision Letter · Decision Letter 1]

18 Feb 2022

Dear Prof. Joshi,

Thank you very much for submitting your revised manuscript "Single domain antibodies against enteric pathogen virulence factors are active as curli fiber fusions on probiotic E. coli Nissle 1917" for consideration at PLOS Pathogens. The revision was reviewed by two of the previous reviewers as well as members of our editorial board. As you can see, the reviewer's assessment of your revised manuscript were mixed. In particular, Reviewer #3 feels that key concerns from the original reviews were not adequately addressed, leaving some of the main conclusions of the paper in doubt. The new reviews are attached below. One outstanding question is whether the modified curli are expressed in the gut or settings that resemble the gut environment. At the very least, the limitations of your study need to be appropriate discussed, which is lacking in the current manuscript. When developing these revisions, please play close attention to the remarks of Reviewer 3. In cases where experimental work was proposed or requested, your response letter should clearly describe either (i) why you feel the work is outside the scope of this manuscript and/or (ii) the technical limitation that would prevent the experiment from being done. 

Sincerely,

Brian K Coombes

Section Editor

PLOS Pathogens

Brian Coombes

Section Editor

PLOS Pathogens

Kasturi Haldar

Editor-in-Chief

PLOS Pathogens

orcid.org/0000-0001-5065-158X

Michael Malim

Editor-in-Chief

PLOS Pathogens

orcid.org/0000-0002-7699-2064

Reviewer Comments (if any, and for reference):

Reviewer's Responses to Questions

**Part I - Summary**

Reviewer #1: Overall the authors addressed the concerns raised.

Reviewer #3: Manuscript by Gelfat et al investigates the engineered VHHs, the variable domains of camelid heavy-chain-only antibodies as ways to inhibit the virulence of multiple pathogenic enteric bacteria. Although this is an exciting study and authors did not address the comments raised by the reviewers. Many comments were left hanging without showing any further data. Authors argue that the some of the information could be found in their previous work. However, the previous studies does not exactly addressing what the reviewers were asking for. Furthermore there are many speculations raised by the authors. Although the reviewers pointed out these concerns were not sufficiently addressed and not discussed as weaknesses of the study or as future directions in the discussion section.

**Part II – Major Issues: Key Experiments Required for Acceptance**

Reviewer #1: (No Response)

Reviewer #3: -There are no structural data shown in the manuscript that the CSG-VHH fibers are assembled into functional entities. Congo red assay that is not specific and one SEM picture is not sufficient to argue that these engineered constructs are working in the way that is proposed in the manuscript. Although SEM pictures are showing the expression of curli, this method is not specific enough to show structural changes. More thorough study of this relationship is not outside the scope of this manuscript. Instead it is needed to confirm that the engineered VHHs are working in the way that they are proposed by the authors. It was suggested in the previous revision to utilize Thioflavin T based amyloid assay supported by TEM or Circular dichroism to confirm the claims done by the authors.

-Authors speculate the curli would be expressed in the intestinal tract but there are no studies that showed that E. coli Nissle expresses curli. The publications that were brought up by the author (Kan, et al. ACS Synth. Biol. 2021, 10, 94−106; Praveschotinunt, et al. Nature Communications, DOI:10.1038/s41467- 019-13336-6) does not directly show that curli is expressed by E. coli Nissle in vivo. The methods used in these previous studies are indirect methods showing E. coli colonization.

-It is necessary to show that at least for one of the virulence factors that the curli-VHH display system would work in vivo to confirm that the system is functional.

-Although Figure S3a shows that, in the absence of CsgA induction there is no fiber formation from PBP8, this doesn’t prove that the VHHs constructs are assembled into right conformation.It only suggest that they are exported and assembled in some fashion. Previous studies shown that CsgA fiber diameters can change in case of wrong assembly. Is there any data to support that the CsgA fiber width is the same in the presence of VHHs? Instead of referring to approximate thickness, measurements can be made as suggested by the reviewers using TEM in the previous submission.Additionally, previously published CsgA fusions were different that the VHH constructs that are being studied in this manuscript. Therefore, it is not relevant to cite publications that used very different structural conformations instead of showing data that would help this manuscript.

**Part III – Minor Issues: Editorial and Data Presentation Modifications**

Reviewer #1: The authors should add the point about the possible misfolding of the VHH to the discussion. The authors should be aware that anti-O or -H antigen should be able to distinguish between EPEC and Nissle (O6:H1).

Reviewer #3: Authors argue that Figure 4C is an idealized schematic that is representative of current theories about curli secretion and assembly that can be found in the literature. Please refer Figure 4C as an idealized schematic as the data is not provided in this manuscript to support that he VHH constructs are assembled in a similar fashion to native CsgA.

PLOS authors have the option to publish the peer review history of their article (what does this mean?). If published, this will include your full peer review and any attached files.

Reviewer #1: No

Reviewer #3: No

Figure Files:

Data Requirements:

Reproducibility:

References:

---

## [Editor Report · Decision Letter 2]

29 Jun 2022

Dear Prof. Joshi,

We are pleased to inform you that your manuscript 'Single domain antibodies against enteric pathogen virulence factors are active as curli fiber fusions on probiotic E. coli Nissle 1917' has been provisionally accepted for publication in PLOS Pathogens.

Best regards,

Matthew A Mulvey, Ph.D.

Associate Editor

PLOS Pathogens

Brian Coombes

Section Editor

PLOS Pathogens

Kasturi Haldar

Editor-in-Chief

PLOS Pathogens

orcid.org/0000-0001-5065-158X

Michael Malim

Editor-in-Chief

PLOS Pathogens

orcid.org/0000-0002-7699-2064
---

## [Editor Report · Acceptance letter]

22 Aug 2022

Dear Prof. Joshi,

We are delighted to inform you that your manuscript, "Single domain antibodies against enteric pathogen virulence factors are active as curli fiber fusions on probiotic E. coli Nissle 1917," has been formally accepted for publication in PLOS Pathogens.

Best regards,

Kasturi Haldar

Editor-in-Chief

PLOS Pathogens

orcid.org/0000-0001-5065-158X

Michael Malim

Editor-in-Chief

PLOS Pathogens

orcid.org/0000-0002-7699-2064